

**1  Bayesian Inference and Predictive Performance of Soil Respiration Models in the Presence**
**2  of Model Discrepancy**

Ahmed S. Elshall[1,2], Ming Ye[3,*], Guo-Yue Niu[4,5] and Greg A. Barron-Gafford[4,6]
[1] Department of Geosciences, University of Hawaii Manoa, Honolulu, Hawaii, USA
[2] Water Resources Research Center, University of Hawaii Manoa, Honolulu, Hawaii, USA
[3] Department of Earth, Ocean, and Atmospheric Science, Florida State University, Tallahassee,
Florida
[4] Biosphere 2, University of Arizona, Tucson, Arizona
[5] Department of Hydrology and Water Resources, University of Arizona, Tucson, Arizona
[6] School of Geography and Development, University of Arizona, Tucson, Arizona
*Corresponding Author: Ming Ye, Telephone: (850) 644-4587, Email: mye@fsu.edu





**Key Points**

(1)    Bayesian inference and prediction are useful to evaluate multiple soil respiration models

with different levels of complexity.

(2)    Data models used in Bayesian inference have substantial impacts on model parameter

distributions and subsequently model predictions.

(3)    Using exponential power distribution and considering heteroscedasticity in data models

improves Bayesian inference and prediction.











Keywords: Soil respiration, Bayesian, likelihood function, data model, autocorrelation,
heteroscedasticity, skew exponential power distribution, cross-validation, relative model score




## Abstract


Bayesian inference of microbial soil respiration models is often based on the assumptions that the
residuals are independent (i.e. no temporal or spatial correlation), identically distributed (i.e.
Gaussian noise) and with constant variance (i.e. homoscedastic). In the presence of model
discrepancy, since no model is perfect, this study shows that these assumptions are generally
invalid in soil respiration modeling such that residuals have high temporal correlation, an
increasing variance with increasing magnitude of $CO_2$ efflux, and non-Gaussian distribution.
Relaxing these three assumptions stepwise results in eight data models. Data models are the basis
of formulating likelihood functions of Bayesian inference. This study presents a systematic and
comprehensive investigation of the impacts data model selection on Bayesian inference and
predictive performance. We use three mechanistic soil respiration models with different levels of
model fidelity (i.e. model discrepancy) with respect to number of carbon pools and explicit
representations of soil moisture controls on carbon degradation, and accordingly have different
levels of model complexity with respect to the number of model parameters. The study shows data
models have substantial impacts on Bayesian inference and predictive performance of the soil
respiration models such that: (i) the level of complexity of the best model is generally justified by
the cross-validation results for different data models; (ii) not accounting for heteroscedasticity and
autocorrelation might not necessarily result in biased parameter estimates or predictions, but will
definitely underestimate uncertainty; (iii) using a non-Gaussian data model improves the parameter
estimates and the predictive performance; and (iv) separate accounting for autocorrelation or joint
inversion of correlation and heteroscedasticity can be problematic and requires special treatment.
Although the conclusions of this study are empirical, the analysis may provide insights for
selecting appropriate data models for soil respiration models.



## 1  Introduction


Developing accurate soil respiration models is important for realistic projection of global
carbon [C] cycle, as global soils store 2,300Pg carbon, an amount more than 3 times that of the
atmosphere (Schmidt et al., 2011) and release 60–75 Pg C/yr, about 7 times more $CO_2$ to the
atmosphere than all human-caused emissions (Le Quéré et al., 2014). The major work on soil
respiration modeling has been focused on advancing knowledge about model inputs and
calibration data (e.g. Janssens et al., 2003; Peters et al., 2007; Scott et al., 2009; Barron-Gafford et
al., 2011; Hilton et al., 2014)  and on developing more advanced models for better representing
soil microbial processes (e.g. Schimel and Weintraub, 2003; Allison et al., 2010; Davidson et al.,
2011; Wieder et al., 2013, 2015; Xu et al., 2014; Zhang et al., 2014) . Integration of data and
models is indispensable for improving predictability of the terrestrial carbon cycle, and statistical
modeling is a vital tool for the model-data integration (Luo et al., 2011, 2014; Wieder et al., 2015).
In addition, use of state-of-the-art statistical methods is necessary to accurately quantify
uncertainty in parameters and structures of soil respiration models for improvement and practical
uses of the models (Katz et al., 2013). Statistical modeling always requires adequately
characterizing residuals, i.e., the difference between data and corresponding model simulations.
While a large number of data models have been used, to our knowledge, comprehensive and
systematic evaluation of data models for soil respiration models has not been reported in literature.
The goal of this study is to evaluate the impacts of data models on Bayesian inference and
predictive performance of three mechanistic soil respiration models, and use these findings to
make broader recommendations. The three models were developed by Zhang et al. (2014) to
simulate the Birch effect (the peak soil microbial respiration pulses in response to episodic rainfall
pulses) at a site scale and a short temporal scale, which are important for gaining mechanistic





understanding of $CO_2$ efflux production (Högberg and Read, 2006; Vargas et al., 2011). Zhang et
al. (2014) developed a total five models, including an existing four-carbon pool model and four
new models with additional carbon pools and/or explicit representations of soil moisture controls
on carbon degradation and microbial uptake rates. The models Zhang et al. (2014) were calibrated,
and Bayesian model selection was used to select and the best model. However, this effort was
based on a single data model. It is unknown whether the best model still remains the best (in terms
of reproducing the both calibration data and the cross-validation data) if a different data model is
used. In addition, since predictive performance of the models was not evaluated in Zhang et al.
(2014), it is unknown whether the best model will give the best predictions. These two questions
are addressed in this study by considering eight data models and by evaluating predictive
performance in a manner of cross-validation. The top two models (also the two most high fidelity
models) ranked by Zhang et al. (2014) are considered in this study, and the worst model (also the
low fidelity model) is also considered in this study for comparison. Model fidelity refers to the
degree of realism of representing our scientific knowledge with respect to the real world system.
That is model with less discrepancy. Conducting Bayesian inference and evaluating predictive
performance for the three models with different degrees of fidelity provides more insights than for
a single model.
Bayesian inference in general uses the Bayes' theorem to update the distributions of model
parameters to posterior parameter distributions given a likelihood function. The mathematical
formulation of the (formal and informal) likelihood function requires a probabilistic data model
that however is intrinsically unknown due to unknown errors in all model components such as
observation data, model structures, parameters, and driving forces. Bayesian inference of soil
respiration models often adopts the assumption of independent, normally distributed and





homoscedastic residuals (e.g. Ahrens et al., 2014; Barr et al., 2013; Hararuk et al., 2014;
Hashimoto et al., 2011; Klemedtsson et al., 2008; Raich et al., 2002; Ren et al., 2013; Ricciuto et
al., 2011; Richardson and Hollinger, 2005; Steinacher and Joos, 2016; Tucker et al., 2014; Tuomi
et al., 2008; Xu et al., 2006; Yeluripati et al., 2009; Zhang et al., 2014; Zhou et al., 2010). These
assumptions are conveniently adopted since the requirement of using an unknown probability
model in Bayesian statistics is called "a basic dilemma" by Box and Tiao (1992). Postulating the
data models is always based on assumptions about residual statistics, and the most widely used
assumptions are paired as follows: (i) independent vs. correlated residuals, (ii) homoscedastic vs.
heteroscedastic residuals, and (iii) Gaussian vs. non-Gaussian residuals.  There are many
diagnostics available to assess these choices (a number of them is used in this paper). However,
few studies have focused on investigating appropriateness of the assumptions for soil respiration
modeling by relaxing the independent residuals assumption (Chevallier and O'Dell, 2013; Ricciuto
et al., 2011) and the Gaussian residuals assumption ( Ricciuto et al., 2011; Van Wijk et al., 2008).
This study evaluates the above assumptions by considering eight data models which relaxes
these three assumptions stepwise as shown in Section 2. For example, combining the assumptions
of independent, homoscedastic, and Gaussian residuals leads to the standard least squares data
model. This model is the simplest one among the eight data models, since it requires only one
parameter, i.e., the constant variance of the Gaussian distribution. Note that there is a difference
between the physical model parameters and data model parameters. They technically can be
estimated together, but one arises from assumptions about process, and the other assumptions
about the data models. Relaxing the homoscedastic assumption to heteroscedastic gives the
weighted least squares data model. It is more complex, because it requires multiple variances for
multiple data. Whenever one or combinations of the three assumptions (independence,



homoscedasticity, and normality) are relaxed, the resulting data models become more complex and
require more parameters. This systematic way of formulating data models is similar to that of
Smith et al. (2010b, 2015), and it is necessary to evaluate appropriateness of the three basic
assumptions and their impacts on Bayesian inference.
The assumptions of heteroscedastic, correlated, and non-Gaussian residuals are accounted for
using the method of Schoups and Vrugt (2010) in the following procedure: (i) the correlation is
removed from the residuals by using an autoregressive model; (ii) the resulting residuals are
normalized by a linear model of variance; and (iii) the normalized residuals are characterized by
using the skew exponential power distribution. The data model parameters (i.e., coefficients of the
autoregressive model, the linear variance model, and the skew exponential power distribution) are
not specified by users, but estimated together with physical model parameters during the Bayesian
inference. The skew exponential power distribution is general in that by adjusting the values of its
kurtosis and skewness parameters the distribution can produce other distributions such as the
Laplace distribution used by (Van Wijk et al., 2008) and (Ricciuto et al., 2011), and other
distributions given by using different kurtosis parameters of an exponential model (Tang and
Zhuang, 2009).  It is worth pointing out that there exist other methods to account for the three
assumptions. Evin et al. (2013) suggested accounting for residual heteroscedasticity before
accounting for residual autocorrelation. Lu et al. (2013) developed an iterative two-stage procedure
to separately estimate physical model parameters and data model parameters. Evin et al. (2014)
developed a similar procedure to first estimate model parameters and then estimate
heteroscedasticity and autocorrelation parameters. While this study uses the method of Schoups
and Vrugt (2010), exploring other methods is warranted in future studies.



After investigating the impacts of the data models on Bayesian inference, this study evaluates
the impacts of the data models on predictive performance of the three soil respiration models.
Using random samples generated during the Bayesian inference, a prediction ensemble is produced
for each soil respiration model. The ensemble is used to evaluate predictive performance of the
models in a stochastic sense by estimating to what extent the models can predict future events. The
evaluation in this study is done in a cross-validation manner to split a dataset of $CO_2$ efflux into
two parts for Bayesian inference and cross-validation, respectively. The evaluation of predictive
performance is important because different data models may give different parameter distributions
and accordingly different predictive performance. For example, the study of van Wijk et al. (2008)
concluded that the choice of the residual function is crucial to achieve accurate model prediction
and parameter estimation. Shi et al. (2014) showed that the posterior parameter distributions and
predictive performance given by two data models (weighted least square and skew exponential
power distribution after removing heteroscedasticity and autocorrelation) are dramatically
different, and a definitive conclusion was drawn that one data model is better than the other. The
evaluation of predictive analysis is conducted for the following two cases: (1) the prediction
ensemble is generated by random samples of the soil respiration models only (i.e. credible
interval), and (2) the prediction ensemble is generated by random samples of not only the soil
respiration models but also the data models (i.e. predictive interval). The two cases lead to different
conclusions about the predictive performance. It is expected that the evaluation of predictive
performance conducted in this study can help select the most appropriate data model to achieve
optimal model predictions.
The remainder of the paper is organized as follows. Section 2 starts with a description of the
evolving data models and their corresponding likelihood functions used in Bayesian inference,

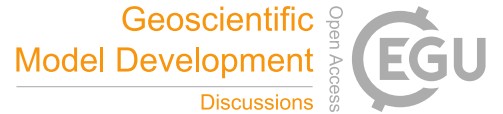



followed by a brief summary of the three soil respiration models. The results of Bayesian inference
are discussed in Section 3 and Section 4, addressing the data model implications on parameter
estimation and predictive performance, respectively. Section 5 summarizes the key findings and
limitations of this study, and provides recommendations for approaching data model selection.
**2    Methodology**
This section starts with a descriptions of the eight data models that account for the three pairs
of assumptions about residuals in a stepwise manner in Section 2.1. The data models are used to
build the likelihood functions used in Section 2.2 for Bayesian inference. The three soil respiration
models and observations of $CO_2$ efflux are described in Sections 2.3 and 2.4, respectively.
**2.1    Data models**
This study considers eight evolving data models starting from a data model that assumes
independent, homoscedastic, and Gaussian residuals to a data model that relaxes all the three
assumptions. The eight data models are based on the generic normalized residual,
$$a_t = \frac{\varepsilon_t}{\sigma_t} \qquad a_t \sim X \,, \tag{1}$$
where $\varepsilon_t = d_t - E_t$ is the residual (the difference between data $d_t$ and its corresponding model
simulation $E_t$) at time or location $t$, $\sigma_t$ is the standard deviation of the residual, and $X$ is the
probability density function (PDF) of $a_t$. The eight data models are formulated with different forms
of $\varepsilon_t$, $\sigma_t$, and $X$. The standard least square (SLS) data model is
$$a_t = \frac{\varepsilon_t}{\sigma_0} \qquad a_t \sim N(0,1) \,, \tag{2}$$
where $\sigma_t = \sigma_0$ is a constant for all the data (i.e., homoscedasty), and $X$ is the standard normal
distribution, $N(0,1)$. The unknown parameter $\sigma_0$ is estimated jointly with unknown physical



model parameters. If $\sigma_t$ is not a constant (i.e., heteroscedasty), SLS becomes the weighted least
squared (WLS) data model. While heteroscedasty can be accounted for through residuals
transformation (e.g. Thiemann et al., 200; Smith et al., 2010b) or other similar approaches (Gragne
et al., 2015) a linear heteroscedastic model $\sigma_t = \sigma_0 + \sigma_1 E_t$ is assumed following other studies
(Thyer et al., 2009; Schoups and Vrugt, 2010; Evin et al., 2013, 2014). With the linear model,
there is no need to estimate $\sigma_t$ for each data. Instead, $\sigma_t$ is calculated by estimating only two
parameters, $\sigma_0$ and $\sigma_1$. The WSL data model is written as
$$a_t = \frac{\varepsilon_t}{\sigma_0 + \sigma_1 E_t} \qquad a_t \sim N(0,1).$$     (3)
The two unknown parameters $\sigma_0$ and $\sigma_1$ are estimated jointly with unknown physical model
parameters. The linear model assigns smaller weight to the data with larger simulation, $E_t$. If the
simulation is small and $\sigma_0 \gg \sigma_1 E_t$, the weight becomes constant for all data. Both SLS and WLS
assume that $a_t$ is independently and identically distributed.
It is not uncommon that residuals are correlated in space and time, due to propagation of
measurement errors (Tiedeman and Green, 2013) and model structure errors (Evin et al., 2014;
Kavetski et al., 2013; Lu et al., 2013). The temporal correlation that occurs in the numerical
example of this study can be accounted for using a *p*-order autoregressive model. This leads to the
data model of standard least square with autocorrelation (SLS-AC),
$$a_t = \frac{\varepsilon_t - \sum_{i=1}^{p} \phi_i \varepsilon_{t-i}}{\sigma_0} \qquad a_t \sim N(0,1)$$     (4)

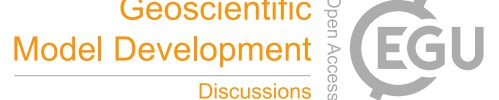



where $p$ is the order of autocorrelation, and $\phi_i$ is an autocorrelation coefficient. The unknown $\phi_i$
and $\sigma_0$ are estimated together with unknown model parameters. By extending the concept of
correlated residuals to WLS leads to the weight least square with autocorrelation (WLS-AC),
$$a_t = \frac{\varepsilon_t - \sum_{i=1}^{p} \phi_i \varepsilon_{t-1}}{\sigma_0 + \sigma_1 E_t} \qquad a_t \sim N(0,1) \tag{5}$$
The unknown parameters of $\sigma_0$, $\sigma_1$, and $\phi_i$ are estimated jointly with physical model
parameters. Equations (2) – (5) assume that the residuals are Gaussian.
The next four data models are similar to the previous four models except that the standard
normal distribution of $a_t$ is replaced by the skew exponential power distribution, $SEP(0,1,\xi,\beta)$,
(Schoups and Vrugt, 2010)
$$p(a_t \mid \xi, \beta) = \frac{2\sigma_\xi}{\xi + \xi^{-1}} \omega_\beta \exp\left[ -c_\beta \left| a_{\xi,t} \right|^{2/(1+\beta)} \right], \tag{6}$$
where zero is mean, one is standard deviation, $\xi$ is skewness, $\beta$ is kurtosis,
$$a_{\xi,t} = (\mu_\xi + \sigma_\xi a_t) \Big/ \xi^{sign(\mu_\xi + \sigma_\xi a_t)} \quad , \quad \mu_\xi = M(\xi - \xi^{-1}) \quad , \quad \omega_\beta = \frac{\Gamma^{1/2}[3(1+\beta)/2]}{(1+\beta)\Gamma^{-3/2}[(1+\beta)/2]} \quad ,$$
$$\sigma_\xi = \sqrt{(1-M^2)(\zeta^2 + \zeta^{-2}) + 2M^2 - 1} \quad , \quad M = \frac{\Gamma[1+\beta]}{\Gamma^{1/2}[3(1+\beta)/2]\Gamma^{1/2}[(1+\beta)/2]} \quad , \quad \text{and}$$
$$c_\beta = \left( \frac{\Gamma[3(1+\beta)/2]}{\Gamma[(1+\beta)/2]} \right)^{1/(1+\beta)} \text{ are derived variables of } \beta \text{ and } \xi, \text{ and } \Gamma[.] \text{ is the gamma function. The}$$
kurtosis parameter $\{\beta \in \mathbb{R} : -1 \le \beta \le 1\}$ determines the peakness of the pdf such that the $\beta$ values
of -1, 0, and 1 give uniform, Gaussian and Laplace distributions, respectively. The skewness
parameter $\{\xi \in \mathbb{R} : 0.1 \le \xi \le 10\}$ determines the skewness of the pdf such that the $\xi$ values of 0.1,
1, and 10 give positively skewed, symmetric, and negatively skewed distributions, respectively.





Setting $\beta = 0$ and $\xi = 1$ leads to $\mu_\xi = 0$, $\sigma_\xi = 1$, $\omega_\beta = 1/\sqrt{2\pi}$, $c_\beta = 1/2$ and $a_{\xi,t} = a_t$, and the
skew exponential power distribution $SEP(0,1,\xi=1,\beta=0)$ becomes the standard normal
distribution,
$$p(a_t \mid \xi = 1, \beta = 0) = \frac{1}{\sqrt{2\pi}} \exp\left[ -\frac{1}{2}(a_t)^2 \right]. \tag{7}$$
which is the data model of SLS in equation (2).
Replacing $a_t \sim N(0,1)$ with $a_t \sim SEP(0,1,\xi,\beta)$ in equations (2) – (5) leads to the data models
SEP, WSEP, SEP-AC, and WSEP-AC as follows,
$$a_t = \frac{\varepsilon_t}{\sigma_0} \qquad a_t \sim SEP(0,1,\xi,\beta) \tag{8}$$
$$a_t = \frac{\varepsilon_t}{\sigma_0 + \sigma_1 E_t} \qquad a_t \sim SEP(0,1,\xi,\beta). \tag{9}$$
$$a_t = \frac{\varepsilon_t - \sum_{i=1}^{p} \phi_i \varepsilon_{t-1}}{\sigma_0} \qquad a_t \sim SEP(0,1,\xi,\beta) \tag{10}$$
$$a_t = \frac{\varepsilon_t - \sum_{i=1}^{p} \phi_i \varepsilon_{t-1}}{\sigma_0 + \sigma_1 E_t} \qquad a_t \sim SEP(0,1,\xi,\beta) \tag{11}$$
In comparison with the Gaussian data models, the SEP-based data models have two more
parameters ($\xi$ and $\beta$) to be estimated jointly with physical model parameters. WSEP-AC data
model, which is known as the generalized likelihood function, is the most commonly used SEP-
based data model (e.g. Vrugt and Ter Braak, 2011; Hublart et al., 2016).
**2.2  Bayesian inference and likelihood functions**
Consider a Bayesian inference problem for a nonlinear model, *f*, used to simulate state
variables (e.g., CO$_2$ efflux), $d = f(\theta) + \varepsilon$, where $d$ is a vector of data, $\theta$ is a vector of model



parameters, and $\boldsymbol{\varepsilon}$ is a vector of residuals that may include errors in data, model parameters, and
model structures. The goal of Bayesian inference is to estimate the posterior distributions, $p(\boldsymbol{\theta}|\boldsymbol{d})$,
of model parameters, $\boldsymbol{\theta}$, given data, $\boldsymbol{d}$, using Bayes' theorem (Box and Tiao, 1992)
$$p(\boldsymbol{\theta}\,|\,\boldsymbol{d}) = \frac{p(\boldsymbol{d}\,|\,\boldsymbol{\theta})\,p(\boldsymbol{\theta})}{\int p(\boldsymbol{d}\,|\,\boldsymbol{\theta})\,p(\boldsymbol{\theta})\,d\boldsymbol{\theta}} \qquad (12)$$

where $p(\boldsymbol{\theta})$ is the prior distribution, and $p(\boldsymbol{d}|\boldsymbol{\theta})$ is the likelihood function to measure goodness-of-
fit between model simulations, $f(\boldsymbol{\theta})$, and data, $\boldsymbol{d}$. The prior distribution can be obtained from data
of previous studies or expert judgment. When prior information is lacking, a common practice is
to assume uniform distributions with relatively large parameter ranges so that the prior
distributions do not affect the estimation of posterior distributions.

The data models above can be used to construct the likelihood functions. For the Gaussian data

models given in equations (2) – (5), the corresponding Gaussian likelihood functions are
straightforward, and an example is equation (7). For the SEP data models, the corresponding
likelihood that is called generalized likelihood function is (Schoups and Vrugt, 2010)
$$p(\boldsymbol{d}\,|\,\boldsymbol{\theta}) = p(\boldsymbol{\varepsilon}_t\,|\,\boldsymbol{\theta}) = \prod_{t=1}^{n} \sigma_t^{-1} \frac{2\sigma_\xi}{\xi + \xi^{-1}} \omega_\beta \exp\left(-c_\beta \left|a_{\xi,t}\right|^{2/(1+\beta)}\right). \qquad (13)$$

where $n$ is the dimension of $\boldsymbol{d}$. The Gaussian likelihood functions are special case of the generalized
likelihood functions. For example, by setting $\beta = 0$, $\xi = 1$, $\phi_i = 0$, $\sigma_t = \sigma_0$, $\sigma_\xi = 1$, $\mu_\xi = 0$,
$\omega_\beta = 1/\sqrt{2\pi}$, $c_\beta = 1/2$, and $a_{\xi,t} = a_t$, equation (13) becomes the likelihood function corresponding
to the SLS data model. Replacing $\sigma_t = \sigma_0$ by $\sigma_t = \sigma_0 + \sigma_1 E_t$, equation (13) becomes the likelihood
function of the WLS data model.

In this study, the distributions of the data model parameters are obtained jointly with the

physical model parameters using the MT-DREAM$_{(ZS)}$ code (Laloy and Vrugt, 2012), which





implements a Markov chain Monte Carlo (MCMC) algorithm by running multiple Markov chains
in parallel with discrete proposal distribution, multiple-try sampling,  and sampling from an
archive of past states. These state-of-the-art features assist in overcoming common challenges in
the sampling landscape such as multimodality, ill-conditioning, and high dimensionality, and thus
allow for accurate exploration of the targeted distributions.
**2.3    Soil respiration models**
Zhang et al. (2014) studied the Birch effect (the peak soil microbial respiration pulses in
response to episodic rainfall pulses), and developed five models, evolving from an existing four-
carbon pool model to models with additional carbon pools and/or explicit representations of soil
moisture controls on carbon degradation and microbial uptake rates. Three of the five models are
used in this study, and they are dented as 4C, 5C, and 6C. Note that model 4C is model 4C_NOSM
of Zhang et al. (2014), not their model 4C. Figure 1 is the diagram of model 6C, the most complex
one among the five models. The simplest one, model 4C, has four carbon pools, i.e., soil organic
carbon (SOC), dissolved organic carbon (DOC), microbial biomass (MIC), and enzymes (ENZ),
and does not consider the soil moisture control on carbon degradation and microbial uptake rates.
Models 5C and 6C has an explicit representation of soil moisture controls on the rates. Based on
the dual Arrhenius and Michaelis–Menten kinetics model, the original SOC degradation rate,
$V_{decom}$, is (Davidson et al., 2011; Davidson and Janssens, 2006)
$$V_{decom} = V_{max} C_{ENZ} \frac{C_{SOC}}{K_m + C_{SOC}}$$    (14)
where $V_{max}$ [s$^{-1}$] is the maximum SOC degradation rate per unit enzyme when the substrates is not
limiting, $C_{ENZ}$ [gCm$^{-3}$] is enzyme pool size, $C_{SOC}$ [gCm$^{-3}$] is SOC pool size, and $K_m$ is the half-

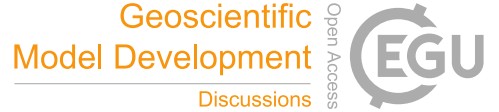



saturation for SOC. The original microbial uptake rate, $V_{uptake}$, is (Davidson et al., 2011; Davidson
and Janssens, 2006)
$$V_{uptake} = V_{max\_up} C_{MIC} \frac{C_{DOC}}{K_{m\_up} + C_{DOC}} \frac{C_{O2}}{K_{m\_upO2} + C_{O2}},$$ (15)
where $V_{max\_up}$ [s$^{-1}$] is the maximum DOC uptake rate when the substrates is not limiting, $C_{MIC}$
[gCm$^{-3}$] is the microbial biomass pool size, $C_{DOC}$ [gCm$^{-3}$] is the DOC pool size, $C_{O2}$ [m$^3$m$^{-3}$] is
the gas concentration of O$_2$ in the soil pore, and $K_{m\_up}$ [gCm$^{-3}$] and $K_{m\_upO2}$ [m$^3$m$^{-3}$] are the
corresponding half-saturation constants for DOC and O$_2$, respectively. With the explicit
representation of soil moisture control, the two rates become (Zhang et al., 2014)
$$V_{decom} = V_{max} C_{ENZ} \frac{C_{SOC}}{K_m + C_{SOC}} \left( \frac{\theta}{\theta_s} \right)$$ (16)
$$V_{uptake} = V_{max\_up} C_{MIC} \frac{C_{DOC}}{K_{m\_up} + C_{DOC}} \frac{C_{O2}}{K_{m\_upO2} + C_{O2}} \left( \frac{\theta}{\theta_s} \right)$$ (17)
where $\theta$ [-] is the volumetric soil moisture, and $\theta_s$ [-] is the porosity.

In addition to using the new rate equations, models 5C and 6C have more carbon pools. In

model 5C, DOC is split into two sub-pools for wet zone and dry zone of soil pores, and only the
wet DOC is used by MIC, as shown in Figure 1. The moisture-controlled microbial uptake rate
becomes
$$V_{uptake} = V_{max\_up} C_{MIC} \frac{C_{DOC\_w}}{K_{m\_up} + C_{DOC\_w}} \frac{C_{O2}}{K_{m\_upO2} + C_{O2}} \left( \frac{\theta}{\theta_s} \right).$$ (18)
where $C_{DOC\_w}$ [gCm$^{-3}$] is the DOC pool size in the wet soil pores. Model 6C is more complex in
that ENZ is further split into two sub-pools for wet and dry pores, and both the wet and dry ENZ



are subject to degradation, as shown in Figure 1. The moisture-controlled SOC degradation rate
becomes

$$V_{decom} = V_{max} C_{ENZ\_W} \frac{C_{SOC}}{K_m + C_{SOC}} \left( \frac{\theta}{\theta_s} \right) \tag{19}$$

for the wet ENZ and

$$V_{decom} = V_{max} C_{ENZ\_D} \frac{C_{SOC}}{K_m + C_{SOC}} \left( 1 - \frac{\theta}{\theta_s} \right) \varepsilon_D \tag{20}$$

for the dry ENZ, where $C_{ENZ\_W}$ [gCm$^{-3}$] is the wet soil pores enzyme pool size, $C_{ENZ\_D}$ [gCm$^{-3}$] is
the enzyme pool size in the dry soil pores, and $\varepsilon_D$ is the catalysis efficiency of the dry zone enzyme.
Due to considering the moisture control and adding more soil pools, model 5C is expected to
be significantly better than model 4C for simulating the Birch effect. Since the accumulated ENZ
in dry soil is secondary, model 6C is expected to be slightly better than model 5C. In terms of
model structural error, model 4C has the largest model structure error, model 5C has significantly
less model structure error, and model 6C has the smallest model structural error. As shown below,
the degree of model structural error is reflected in the process of Bayesian inference and verified
by the cross-validation.
**2.4    Observations and parameter estimation**
Figure 2 plots the time series of 17,016 observations of soil moister and $CO_2$ efflux used in
this study. The observations were obtained during the entire year of 2007, covering a long period
of dry season prior to monsoon and episodic rainfall events during monsoon. The first two third of
this dataset is used for the Bayesian inference, and the last one third is used for cross-validation.
The inference and cross-validation periods have both dry and wet periods, as shown in Figure 2.
The observation site is located within the Santa Rita Experimental Range (SRER, 31.8214°N,





110.8661°W, elevation 1,116 m) outside of Tucson, Arizona (Barron-Gafford et al., 2011; Scott
et al., 2009). This savanna site was covered by 22% of perennial grass, forbs and subshrubs and
35% of mesquite. The soils are uniformly Comoro loamy sand (77.6% sand, 11.0% clay, and
11.4% silt). The half-hourly atmospheric forcing data were collected from measurements through
an eddy covariance tower (Scott et al., 2009). This includes downward shortwave, longwave,
precipitation, wind, air temperature, humidity, and pressure. Volumetric $CO_2$ concentration was
measured at half-hourly interval through compact probes. The $CO_2$ efflux was estimated from the
gradient of $CO_2$ concentration measured at two depths of 2 cm and 10 cm through Fick's first law
of diffusion, and the estimates were validated against measurements from a portable $CO_2$ gas
analyzer.

The parameters estimated in this study include the parameters of the soil respiration models

(4C – 6C) and the parameters of the data models described in Section 2.1. The estimated
parameters of models 4C and 5C include the microbial carbon use efficiency (CUE) [g/g], enzyme
production rate, $k_e$ [g/m$^3$s], microbial turnover rate, $\tau_m$ [1/s], and enzyme turnover rate $\tau_e$ [1/s].
Uniform distributions are used as the prior in the Bayesian inference, and the ranges of the four
parameters are 0.2 – 1.00, $1\times10^{-12} – 1\times10^{-7}$, $1\times10^{-12} – 1\times10^{-5}$ and $1\times10^{-11} – 1\times10^{-6}$, respectively.
The values of other parameters are fixed at the values used in Allison et al. (2010). Model 6C has
two more parameters, and they are the catalysis efficiency $\varepsilon_D$ [-] and the turnover rate of the dry-
zone enzymes $\tau_{en}$ [1/s]. The prior of the two parameters are uniform distributions with the ranges
of 0.2 – 0.8 and $1\times10^{-12} – 1\times10^{-8}$, respectively.

The DREAM-based MCMC simulation is conducted for a total of 24 cases, the combinations

of eight data models and three physical models. For each case, the parameter distributions are
obtained after drawing a total of $5\times10^5$ samples using five Markov chains. The Gelman and Rubin





(1992) R-statistic is used for convergence diagnostic, and it approaches one in less than $4 \times 10^4$
samples. The initial 50% of the samples are discarded during the burn-in period.

### 3    Results of Bayesian Inverse Modeling

This section analyzes the residuals of the best realization (with the highest likelihood value) of
the MCMC simulation to understand whether the assumptions of the eight data models hold. The
impacts of the data models on the posterior parameter distributions are also analyzed.

### 3.1    Residual characterization

Figure 3 shows residual plots for model 6C based on data models SLS and WSEP-AC. SLS is
the simplest one with the assumptions of homoscedastic, independent, and Gaussian residuals, and
the WSEP-AC is the most complex one without the assumptions. Model 6C is the most complex
model and also the best one as ranked by Zhang et al. (2014) using Bayesian model selection. The
variable $a_t$ plotted in Figures 3a-3c and Figures 3d-3f is defined in equations (2) and (11),
respectively. Figures 3a – 3c show that the three residual assumptions are violated when SLS is
used because (i) the residual variance is not constant, but increases as a function of the simulated
$CO_2$ efflux (Figure 3a); (ii) the autocorrelation function at most lags is beyond the 95% confidence
interval (Figure 3b); (iii) and the standard normal density function cannot adequately characterize
the residuals (Figure 3c). Figures 3d-f show that, after relaxing the three assumptions, the
processed residuals, $a_t$, can be well characterized by WSEP-AC. Figure 3d shows that, after
normalizing $\varepsilon_t$ with the linear variance ( $\sigma_t = 0.034 + 0.099 E_t$ ), the variation of the variance of
$a_t$ becomes significantly smaller, although the variance is still not a constant. Figure 3e shows that,
after removing a first-order autoregressive model from $\varepsilon_t$, $a_t$ becomes less correlated, although the
correlation is not fully removed. The two coefficients of the autoregressive model are $\phi_1 = 0.989$
and $\phi_2 = 4.5 \times 10^{-6}$; the small value of $\phi_2$ indicates that there is no need to attempt an autoregressive





model of higher order. Figure 3f shows that $a_t$ follows the SEP distribution with the estimated
skewness coefficient of $\xi = 0.933$ and kurtosis coefficient of $\beta = 0.998$. As a summary, Figure
3 shows that it is important to examine the residuals and to determine whether a data model is
adequate for charactering the residuals. Although WSEP-AC still cannot perfectly characterize $\varepsilon_t$,
it is significantly better than SLS.
Although the Gaussian assumption used in SLS is violated for model 4C (Figure 3c), this is
not generally the case for other data models and physical models. This is shown in Figure 4, which
presents the quantile-quantile (Q-Q) plot for the eight data models and the three soil respiration
models. For SLS, WLS, SLS-AC, and WLS-AC, the theoretical quantiles are based on the standard
normal distribution, $N(0,1)$; for SEP, WSEP, SEP-AC, and WSEP-AC, the theoretical quantiles
are based on the standard skew exponential power distribution, $SEP(0,1,1,0)$. If the residuals
follow the assumed standard distributions, the Q-Q plots fall on the 1:1 line, which is marked as
the theoretical lines in Figure 4. If the residuals are Gaussian or SEP but not standard, the Q-Q
plots fall on a straight line but not the 1:1 line. Figures 4a and 4e show that, for all the soil
respiration models, the Q-Q plots of SLS and SEP deviate significantly from the theoretical lines
and exhibit fat-tail behaviors, which is an indication of outliers (Thyer et al., 2009). The deviation
is reduced after accounting for autocorrelation in SLS-AC and SEP-AC, as shown in Figures 4c
and 4g (it is interesting to observe from the two figures that the Q-Q plots of the three models are
almost visually identical). The deviation is almost fully removed after accounting for
heteroscedasticity in WLS and WSEP in that their corresponding Q-Q plots fall on the 1:1 lines,
especially for models 5C and 6C, as shown in Figures 4b and 4f. However, the Q-Q plots start
deviating from the 1:1 lines as shown in Figures 4d and 4h, after accounting for both
heteroscedasticity and autocorrelation in WLS-AC and WSEP-AC. As a summary, Figure 4 shows



that, for the numerical example of this study, either the Gaussian or the SEP distribution is valid if
heteroscedasticity is accounted for in the data models. However, accounting for autocorrelation in
the data models does not help improve the characterization of the residual distribution.
**3.2      Posterior parameter distributions**

While Figures 3 and 4 help understand validity of the three assumptions used in the data

models, the impacts of the data models on estimating model parameter distributions must be
evaluated separately. This section discusses the impact of the data model selection on parameter
estimation with the objective of understanding if incorrect specification of the data model, will
necessarily lead to biased parameter estimates. Such assessment is not a trivial task for three main
reasons. First, microbial soil respiration models aggregate complex natural processes and spatial
details into simpler conceptual representations. As a results several model parameters are effective
values of several complex natural processes that cannot be actually measured in the field as
discussed by Vrugt et al. (2013). Second, even for model parameter that can be measured in the
field, since the model structure is imperfect, it can be the case that parameter values can be
accepted beyond their physically reasonable range as discussed by Pappenberger and Beven
(2006). This is often undesirable, if we seek to make the models more mechanistically descriptive.

We focus our discussion on carbon use efficiency (CUE) for microbial growth since CUE is a

fundamental parameter in microbial soil respiration models, and a reasonable physical range for
CUE can estimated.  The concept of microbial CUE(Allison et al., 2010; Bradford et al., 2008;
Manzoni et al., 2012; Wieder et al., 2013) has been used to present fundamental microbial
processes recent microbial enzyme models(Allison et al., 2010; German et al., 2011; Schimel and
Weintraub, 2003; Wang et al., 2013). The microbial CUE, which is marked between MIC and CO2
in Figure 1, controls microbial growth, enzyme production and microbial respiration. A reasonable



range of CUE can be estimated from the physical viewpoint(Tang and Riley, 2014). Sinsabaugh
et al. (2013) study shows that the thermodynamic calculations support a maximum CUE of 0.60
and that methods used to estimate CUE in terrestrial systems report a mean value of 0.55.
Theoretically, there no lower limit for CUE as it can approach zero, and CUE< 0.1 are reported
for terrestrial ecosystems (e.g. Fernández-Martínez et al., 2014) and used in modeling studies (Li
et al., 2014).

Figure 5 plots the CUE posterior marginal density of the three soil respiration models obtained

using the eight data models. The physical range between zero and 0.6 is marked in yellow. Figure
5 shows that the CUE posterior parameter distribution for Model 6C for all likelihood functions
that does not account for autocorrelation are within a reasonable physical range. For models 4C
and 5C, the posterior parameter samples are outside the physical range for six data models. For
model 4C, the posterior parameters are within the physical range only for data models SEP and
WSEP; for model 5C, the two data models are WLS and WSEP. It is not surprising to find the
posterior parameter distribution of models 4C and 5C, which have a certain degree of model
structure error, to be out of the plausible physical range. This can be attributed to two reasons.
First, the model solution can be biased toward the missing processes in the model structure such
as the additional carbon pool in both 4C and 5C or the explicit accounting for soil moister in 4C.
Second, biased parameter estimation can compensate for model structure inadequacy and other
sources of discrepancy in both the physical model and the statistical model.

In addition, it is important to understand how accounting for autocorrelation, heteroscedasty

and non-Gaussian residuals can affect the parameter estimation.  First, it is not unexpected to get
biased parameter estimates that can be out the reasonable physical range when autocorrelation is
explicitly accounted for as shown in Figure 5e-h. This may suggest again that accounting for





heteroscedasticity is desirable but accounting for autocorrelation is not. A possible reason is that
filtering autocorrelation may reduce the residual space such that the transformed residual space
cannot correspond to the parameter space of the models. In other words, parameter information
may be lost due to filtering out autocorrelation. However, it is not fully understood why this does
not occur for the model 6C under data model SLS-AC, and more research is warranted.  Second,
unlike accounting for auto-correlation, accounting only for heteroscedasty (i.e. WLS and WSEP)
since this will only amplify or reduce the variance without affecting the structure of the residual
space. Figure 5c-d shows that account for heteroscedasty (i.e. WLS and WSEP) tends to improve
the parameter estimation in comparison with homoscedastic data models (i.e. SLS and SEP) shown
in Figure 5a-b. Finally, with respect to non-Gaussian residuals, Schoups and Vrugt (2010)
proposes that the peaked pdf of the SEP with heavier tails compared to Gaussian pdf is useful for
making parameter inference robust against outliers. To a certain degree, this can be substantiated
by the results in Figure 5a-d, such that SEP and WSEP provide more favorable parameter estimates
than SLS and WLS.
Finally, from Figure 5 we can also notice that the posterior parameter distribution of SLS
(Figure 5a) is very narrow. This narrow posterior parameter distribution of SLS compared to other
likelihood functions can be attributed to several reasons. Since SEP can have heavier tails than
Gaussian distribution, this can further increase the samples acceptance ratio from tails resulting in
wider distribution (Figure 5b). In addition, accounting for heteroscedasticity will wider the
posterior parameter distribution (Figure 5c) due to accepting higher variances at peak effluxes.
Moreover, filtering correlation (Figure 5e-h) increases the entropy.





**4.      Results of Predictive Performance**
Based on the last one third of the $CO_2$ efflux observations, a cross-validation test was conducted
for all the 24 models,  the combinations of three soil respiration models and eight data models.
Given the cross-validation data, the predictive performance is examined using the four statistical
metrics defined in Section 4.1. The metrics are also calculated for the calibration data. This is not
to perform Bayesian model selection given the calibration data, but to better understand the impact
of data models. For each calibration and each cross-validation data, a prediction ensemble is
generated from the two perspectives of parametric uncertainty only and total uncertainty, as
presented in Section 4.2 and 4.3, respectively.
**4.1      Metrics for evaluating predictive performance**
Three criteria are used to evaluate the predictive performance of the soil respiration models
and data models, and they are central mean tendency, dispersion, and reliability. Each criteria is
measured by a single metric. In addition, a newly defined metric is also used for simultaneously
measuring the three criteria. The central mean tendency is measured in this study using the Nash-
Sutcliffe model efficiency (NSME) coefficient (Nash and Sutcliffe, 1970),
$$NSME = 1 - \sum_{i=1}^{n}(d_i - \overline{X_i})^2 \bigg/ \sum_{i=1}^{n}(d_i - \overline{\mathbf{d}})^2 , \qquad (21)$$
where $n$ is the number of cross-validation data, $d_i$ is the $i$-th data, $\overline{\mathbf{d}}$ is the mean of the data, and
$\overline{X_i}$ is the mean of the prediction ensemble, $X_i$, for $d_i$. NSME ranges from -∞ to 1, with $NSME = 1$
corresponding to a perfect match between data and mean prediction, i.e., the ensemble is centered
on the data. $NSME = 0$ indicates that the model predictions are as only accurate as the mean of the
data, while an efficiency $NSME < 1$ indicates that the mean of data is a better prediction than the
mean prediction.





In addition to the central mean tendency, it is also desirable that the ensemble is precise with
small dispersion and reliable to cover all the data. This study uses a nonparametric metric for
dispersion, and it is the sharpness of a prediction interval (e.g. Smith et al., 2010a)
$$Sharpness = 1/n \sum_{i=1}^{n} [Max(X_i) - Min(X_i)] \qquad (22)$$
where $X_i$ is the prediction ensemble within the 95% prediction interval (the Bayesian credible
interval, not the confidence interval used in nonlinear regression (Lu et al., 2013). Smaller values
of sharpness indicate better prediction precision. Reliability is measured using predictive coverage.
(e.g. Hoeting et al., 1999), which is the percentages of data contained in the prediction interval.
Larger predictive coverage values are preferred.
To account for the trade-off between the three metrics,(Elshall et al., 2018)  defined relative
model score (RMS) that simultaneously measure all the three criteria. Scoring rules are commonly
used in hydrology to assess predictive performance (e.g. Weijs et al., 2010; Westerberg et al.,
2011). RMS is used in this study to measure the relative predictive performance of the
combinations of soil respiration models and data models. For combination $M_j$, RMS is defined as
$$RMS(M_j) = \sum_{i=1}^{n} \frac{p(d_i \mid X_{ij}, M_j)}{\sum_{j=1}^{m} p(d_i \mid X_{ij}, M_j)} \times 100 \qquad (23)$$
where $m = 24$ is the number of combinations, and $X_{ij}$ is similar to $X_i$ above and specific to the $j$-th
combination. The density function, $p(d_i|X_{ij})$, can be evaluated by first obtaining the density function
$p(X_{ij})$ of the ensemble prediction $X_{ij}$ (e.g., by using the kernel density function) and then evaluating
$p(d_i|X_{ij})$ using interpolation methods based on the intersection of $X_{ij}$ and $d_i$. This evaluation is based
purely on the model predictions, and does not involve any assumptions on the models, their
parameters, and likelihood functions. Larger RMS values indicate better overall predictive
performance.



### 4.2     Predictive performance with parametric uncertainty of soil respiration models

In this section the ensemble is generated by running the soil respiration models with the

posterior samples (obtained from the Bayesian inference) of the physical model parameters. In

other words, the ensemble addresses parametric uncertainty of the soil respiration models only.

Considering the relative contribution of parametric uncertainty only will provide insights for

modeling approaches that attempt to segregate various sources of uncertainty (e.g. Thyer et al.,

2009; Elshall and Tsai, 2014).

The four statistics above (i.e. NSME, sharpness, coverage, and RMS) are calculated for the

three soil respiration models and the eight data models. Taking data models SLS and WSEP-AC

as an example, Figure 6 plots the data (for the calibration and cross-validation periods separately)

along with the mean and 95% credible intervals of the prediction ensemble for the three models.

Figure 6 shows that the data models affect model simulations for all the models. The statistics,

especially RMS, indicate that WSEP-AC has better predictive performance than SLS. This is most

visually obvious for model 6C during the cross-validation period after 330 days, as the prediction

ensemble of SLS (Figure 6k) cannot cover the observations unlike the prediction ensemble of

WSEP-AC can (Figure 6l). This conclusion that WSEP-AC outperforms SLS agrees with that

drawn from Figures 3 and 4.

Figure 7 plots the four statistics for all the soil respiration models and data models. Figures 7a

and 7b show the predictive performance with respect to the central mean tendency using NSME

for both the calibration and cross-validation periods respectively. The results indicates that the

low fidelity model 4C under all data models will over-fit the data resulting in biased predictions

such that the NSME values become significantly worse (from 0.6 to -0.6) from the calibration to

the cross-validation period. This is confirmed by the visual inspection of Figures 6a, 6b, 6g, and



6h for data models SLS and WSEP-AC. For models 5C and 6C, their NSME values vary with the
data models with the central mean accuracy being the worst for SLS-AC which considers only
autocorrelation.

With respect to parametric uncertainty estimation, Figures 7c and 7d show sharpness generally

increases when the three assumptions in the data models are gradually relaxed from SLS to WSEP-
AC. This is even more obvious during the validation period. Given that the prediction ensemble
does not center on the data, the increasing sharpness is desirable as it improves reliability. This is
confirmed by the reliability plots in Figures 7e and 7f. The exceptions are again SLS-AC and SEP-
AC that generally have the lowest coverage.

With respect to the overall predictive performance, the same variation pattern and exception

are also observed in the RMS plots in Figures 7g and 7h. This is not surprising because RMS is
the metric that can be used to measure all the three criteria (central mean tendency, sharpness, and
reliability). Since the prediction ensemble is not centered on the data, the sharpness and reliability
are the decisive factors for evaluating the predictive performance.

As a summary, while it is necessary to account for heteroscedasticity in a data model, caution

is needed when accounting for autocorrelation in the manner described in Section 2.1. In addition,
after comparing the RMS values of the residuals using the Gaussian and SEP distributions. The
conclusion is that the SEP distribution outperforms the Gaussian distribution with respect to
predictive performance. Finally, uncertainty underestimation as evident by the very small
predictive coverage. The underestimation of uncertainty for all the physical models with all
likelihood function make sense because only parametric uncertainty is considered.  Considering
the overall predictive uncertainty is the subject of the next section.



### 4.3 Predictive performance with parametric uncertainty of soil respiration models and data models

The simulated output $\mathbf{Y}(\theta_p)$ will generally not be equally to the observed output $\mathbf{D}$ and we have a residue error term $\mathbf{e}$ due to measurement, input and model structure errors such that $\mathbf{D} = \mathbf{Y}(\theta_p) + \mathbf{e}$. Accounting for error term $\mathbf{e}$ can be through separating various error terms. For example, in section 4.2 we obtained uncertainty due to the physical model parameters. Accounting for other sources of uncertainty can be done using a single model approach (e.g. Thyer et al., 2009) or a multimodel approach (e.g. Tsai and Elshall, 2013). Alternatively, we can quantify the uncertainty based on total residuals, which include measurement, model input, model structure and parameter estimation errors (e.g. Thyer et al., 2009; Schoups and Vrugt, 2010). This lumped approach is based on sampling the residual error model $\mathbf{e}(\theta_e)$ with parameters $\theta_e$. SLS has one fixed parameter that is the constant variance and other data models have two to six parameters. Thus in Section 4.3, the prediction ensemble addresses parametric uncertainty of not only the soil respiration models but also the data models. When generating the prediction ensemble in the procedure described by Schoups and Vrugt (2010), an ensemble of residuals is first generated by running the data models with posterior samples of the data model parameters for the positive carbon efflux domain; the residual ensemble is then added to the prediction ensemble generated in Section 4.2.

We start by the visual assessment of the predictive performance. Figure 8 is similar to Figure 6 with the exception that Figure 8 considers the overall all predictive uncertainty (i.e. parametric and output uncertainty), while Figure 6 considers the parametric uncertainty only. Figure 8 reveals a practical observation about accounting for the overall uncertainty through the lumped approach of sampling the residual errors model. Figure 8b shows that desp8ite the wide prediction interval



of model 4C, which has significant model structure error, it could not capture the birch pulse
around day 180. This clearly indicates that proper modeling of the residual error will not make-up
for of significant model structure error.

Figure 9 plots the four statistics (NSME, sharpness, predictive coverage, and RMS) of the three

models under the eight data models to assess the predictive performance. First with respect to
central mean tendency, The NSME values in Figures 9a-9b are visually the same as those in
Figures 7a-7b, indicating that the central mean accuracy under parametric uncertainty is the same
as that under predictive uncertainty.

With respect to uncertainty, the values of sharpness and predictive coverage increase

substantially (Figures 9c – 9f). In particular, Figures 9e and 9f show that, except for SLS and SEP,
the predictive coverage of the rest six data models are close to 100% for all the three models,
indicating that the prediction intervals cover almost all the data. This is demonstrated in Figures 6
for WSEP-AC. Similar to Figures 7c and 7d, Figures 9c and 9d also show a general pattern that
the sharpness increases when the three assumptions in the data models are gradually relaxed from
SLS to WSEP-AC. The data models that account for autocorrelation are still the exceptions.

With respect to the overall predictive performance, the RMS values are largely determined by

mean accuracy and sharpness as the predictive coverage is similar for different data models.
Figures 9g and 9h of RMS show that the predictive performance of the four data models that
account for autocorrelation is worse than that of the other four data models. This suggests again
that one needs to be cautious when building autocorrelation into a data model. This is consistent
with the finding of Evin et al. (2013, 2014) that accounting for autocorrelation before accounting
for heteroscedasticity or jointly accounting for autocorrelation and heteroscedasticity can result in
poor predictive performance. In summary, Figures 9g and 9h show for both the calibration and





prediction periods that accounting for heteroscedasticity (i.e. WLS and WSEP) will give the best
overall predictive skill, and accounting for autocorrelation without heteroscedasticity (i.e. SLS-
AC and SEP-AC) will give the worst overall predictive skill. Finally, for the three soil respiration
models, RMS shows that model 4C has the worst predictive performance for both the calibration
and cross-validation data. Generally speaking, the high fidelity model 6C outperforms model 5C
for both the calibration and cross-validation data, which justifies the complexity of model 6C.

To demonstrate the impacts of the data models on predictive performance of the soil respiration

models, Figure 10 plots the model simulations and predictions given by model 6C during the
calibration and cross-validation periods using all the eight data models.

In Figure 10 we try to understand the predictive performance characteristics of the different

data models by looking at the predictive performance of model 6C. Specific predictive
performance patterns can be identified. Figures 10-a-d show that SLS and SEP have similar
predictive performance with SEP generally having better predictive skill especially during the
validation period.  Accounting for heteroscedasticity using WLS as shown in Figures 10e and 10h
will make the predictions more sensitive to peck carbon effluxes and will generally improve the
predictive coverage on the expense of sharpness and the central mean tendency.  WLS and WSEP
have similar predictive performance. However, WSEP maintains slightly better central mean
tendency and overall predictive performance than WLS. Accounting for autocorrelation using
SLS-AC and SEP-AC as shown in Figures 10i and 10l reduces the information content of the
residuals thus resulting in wider uncertainty bands and insensitivity to peak carbon effluxes as
compared to SLS and SEP (Figures 10a-d). This resulted in deteriorating the sharpness, the central
mean tendency and the capturing of peak carbon fluxes, especially during the validation period.
Accounting for both heteroscedasticity and autocorrelation using WLS-AC and WSEP-AC will



make the inference robust against peck carbon effluxes, yet due to the loss of information content
uncertainty bands are still wider and uncertainty becomes overestimated especially during
validation period as compared to WLS and WSEP. The results of Models 4C and 5C, which are
not shown here, also show the same prediction patterns with respect to non-Gaussian residuals,
heteroscedasticity and autocorrelation.

From figure 10 we also notice that data models that have good overall predictive performance

as measured by RMS during the calibration period will maintain this good predictive performance
during the validation period. For model 6C, RMS values for the calibration and validation periods
are very well correlated with a correlation coefficient of 0.92. However, we note that for models
4C and 5C the overall predictive performance during the calibration and validation periods are not
that well correlated as 6C, with correlation coefficients of 0.52 for model 4C and 0.61 for model
5C. This suggests that model 6C is more robust than 4C and 5C for forecasting and hindcasting.
**5.      Conclusions**

In parameter estimation and prediction of soil carbon fluxes to the atmosphere we often

assume that residuals, which include observation, model input, model structure and parameter
estimation errors, are normally distributed, homoscedastic and uncorrelated. We studied these
assumptions by calibrating three microbial enzyme models, which have varying degrees of model
structure errors. We tested eight data modeling starting with the standard least squares (SLS) and
skew exponential power (SEP) data models that assume homoscedasictic and non-correlated
residuals. Given these two distributions, we evaluated six other data models that account for
heteroscedasicty (WLS and WSEP), autocorrelation (SLS-AC and SEP-AC) and joint inversion of
heteroscedasicty and autocorrelation (WLS-AC and WSEP-AC). To our knowledge this is the first
study that provide such detailed analysis soil reparation inverse modeling. We also used three solid



respiration models with different degrees of model realism and model complexity (i.e. number of
model parameters), to understand the impact of model discrepancy on the calibration results under
different data models. We analyzed the calibration results with respect to (i) residual
characterization, (ii) parameter estimation, (iii) predictive performance and (iv) impact of model
discrepancy. The main findings of this study can be summarized as follows:
(i) With respect to residual characterization, residual analysis results suggest that the common
assumption of not accounting for heteroscedasicty and autocorrelation of residuals (i.e. SLS and
SEP) results in poor characterization of residuals. Explicit accounting for heteroscedasicty (i.e.
WLS and WSEP) can result in good characterization of the residuals, and is followed by joint the
inversion of heteroscedasicty and autocorrelation (i.e. WSL-AC and WSEP-AC). Accounting for
autocorrelation only (i.e. SLS-AC and SEP-AC) may not improve much the characterization of the
residuals.
(ii) With respect to parameter estimation, we focused on carbon use efficiency (CUE), which
is a central parameter in soil respiration modeling. We found the SLS with relatively reasonable
posterior parameter distribution for CUE, yet very narrow posterior. Data models consider
autocorrelation (i.e. SLS-AC, SEP-AC, WLS-AC and WSEP-AC) tend to generally yield CUE
estimates that are physically non-reasonable. We speculate that filtering correlation can affect the
mapping of the model physics (as implicitly included in the residuals) into the likelihood space,
which might result in biased parameter estimates that are physically unreasonable.
(iii) With respect to predictive performance, we assessed the central mean tendency,
uncertainty bands and the overall predictive performance for both the calibration and the cross-
validation periods. Results show that accounting for autocorrelation (i.e. SLS-AC, SEP-AC, WLS-
AC, and WSEP-AC) deteriorate the predicative performance, such that the predictive performance



is inferior to SLS in terms of the central mean tendency and overall predictive skill, especially
during the cross-validation period. Results also indicates that using a SEP distribution can
potentially improve the predictive performance. The same is true for accounting for
heteroscedasticity. Using SEP distribution and accounting for heteroscedasticity (i.e. WSEP) can
potentially improve the predictive performance.
(iv) With respect to the impact of model discrepancy, the high fidelity complex model (6C)
gives the best results with respect to parameter estimation and predictive performance. Model 6C
generally maintained its superior performance under different data models. This justifies the
complexity of model 6C relative to model 5C that has one less carbon pool. Model 4C that has a
low fidelity model with only four carbon pools and lacks the explicit representation of soil moisture
control, maintains its poor performance for different data models.
From the empirical findings of this research we conclude the following: (i) Not accounting for
heteroscedasticity and autocorrelation using a Gaussian or non-Gaussian data model might not
necessarily result in biased parameter estimates or biased predictions with respect to central mean
tendency, but will definitely underestimate uncertainty resulting in lower overall predictive
performance. (ii) Using a non-Gaussian residual error model can improve the parameter estimates,
and the predictive performance with respect to central mean tendency and uncertainty estimation.
(iii) Accounting for heteroscedasticity will definitely improve the uncertainty estimation with
respect to reliability at the cost of having a wider predictive interval. (iv) This study confirms the
empirical findings and theoretical analysis of Evin et al. (2013; 2014) that separate accounting for
autocorrelation or joint inversion of correlation and heteroscedasticity can be problematic.
Relatively poor performance with respect to autocorrelation can be due to our implementation
scheme, which can be improved by using the post-processing inference approach for



autocorrelation (Evin et al., 2013; 2014) or similar strategies (Li et al., 2015, 2016). Further
investigation of this point is warranted in a future study.
The conclusions above are subject to several limitations. First, the conclusions are specific to
the soil respiration models developed and validated for semi-arid savannah. Performance
variations across different soil respiration models with different levels of complexities is possible.
Second, the conclusions are conditioned on the data that were obtained at the half-hour interval
over a one-year period. Different conclusions are possible if the data are thinned to daily or weekly
scales or data of longer observation periods are used. Third, the study investigates effects of the
residual assumptions of formal likelihood functions through direct conditioning of the error model
parameters, yet this can also be done through other approaches such as residuals transformation
(Thiemann et al., 2001), autorgressive bias model (Del Giudice et al., 2013), approximate Bayesian
computation (Sadegh and Vrugt, 2013), data assimliation (Spaaks and Bouten, 2013). Comparing
different methods for accounting the residual assumptions are beyond the scope of this work.
Fourth, this study focuses on formal Bayesian computation using formal likelihood functions, and
comparison with other inference functions such as informal likelihood functions or approximate
Bayesian computation is warranted in a future study.
Based on the aforesaid conclusions and limitations, we recommend to start calibrating soil
respiration models with simple SLS or SEP likelihood function. If the residuals characterization is
adequate (e.g. Scharnagl et al., 2011), then the underlying assumptions are met. Otherwise,
increase complexity of the data model until satisfactory results are obtained in terms of residuals
characterization, posterior parameter estimation and predictive performance. Although the
empirical findings of this study provide general guidelines for data model selection of microbial





soil respiration models, more comparative studies are needed to validate and refute the findings of
this study.

**Code and data availability**

The data and codes and models used to produce this paper are available on contact of the
corresponding author at mye@fsu.edu. We cannot publicly share the workflow because MT-
DREAM$_{(ZS)}$ code (Laloy and Vrugt, 2012) , which is a main component in the workflow, is in the
process of becoming a commercial code.

**Author contributions**

ASE developed and implemented the code for the eight data models for soil respiration modeling,
and prepared the manuscript with contribution of all co-authors. MY developed the research idea
and outline, and supervised the research implementation. GN developed the soil respiration
models. GAB collected and processed the eddy-covariance data used for model calibration.

**Competing interests**

The authors declare that they have no conflict of interest.

**Acknowledgement**

This work was supported by the Department of Energy Early Career Award, DE-SC0008272 and
U.S. National Science Foundation Award# OIA-1557349.

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



Figure 1. Diagram of model 6C representing the processes of (1) degradation of soil organic carbon
(SOC) to dissolved organic carbon (DOC) through catalysis of enzymes (ENZ) produced by
microbes (MIC), (2) MIC uptake of DOC, and (3) microbial (MIC) respiration to produce $CO_2$
(CUE is the carbon use efficiency). SOC degradation and microbial uptake rates are controlled by
water saturation $(\theta/\theta_s)$. The DOC and ENZ pools are split into two subpools, one for the wet zone
and the other for the dry zone of the soil pore space. Microbial uptake of DOC occurs only in the
wet zone, and the uptake rate is linearly related to $\theta/\theta_s$. Catalysis through ENZ in the wet zone is
proportional to $\theta/\theta_s$, while that in the dry zone is proportional to $1-\theta/\theta_s$. $V_{max}$ (s$^{-1}$) is the maximum
rate, and $K_m$ is the half-saturation concentration.

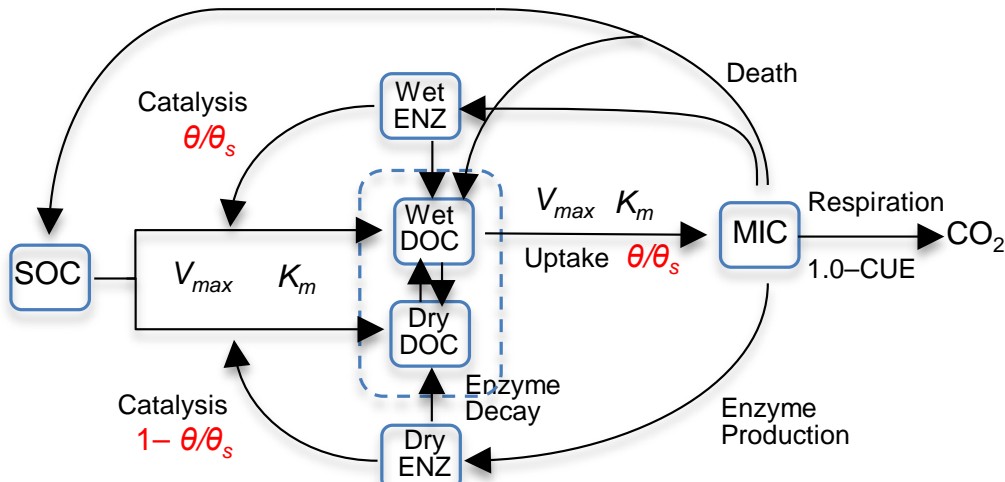






Figure 2. Time series of soil moisture and efflux observations. The dashed line marks the divide
of the dataset into calibration and validation periods.

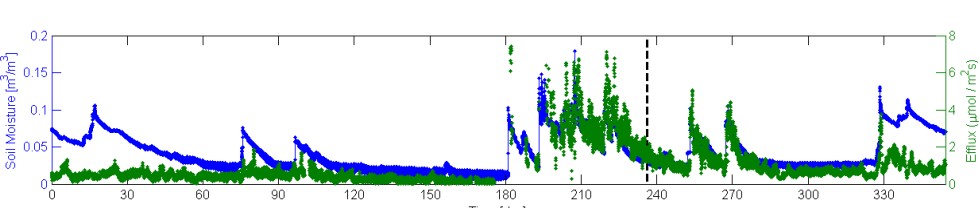






Figure 3. Residual analysis of the best realization (among multiple MCMC realizations) for model
6C using data models (a-c) SLS and (d-f) WSEP-AC.

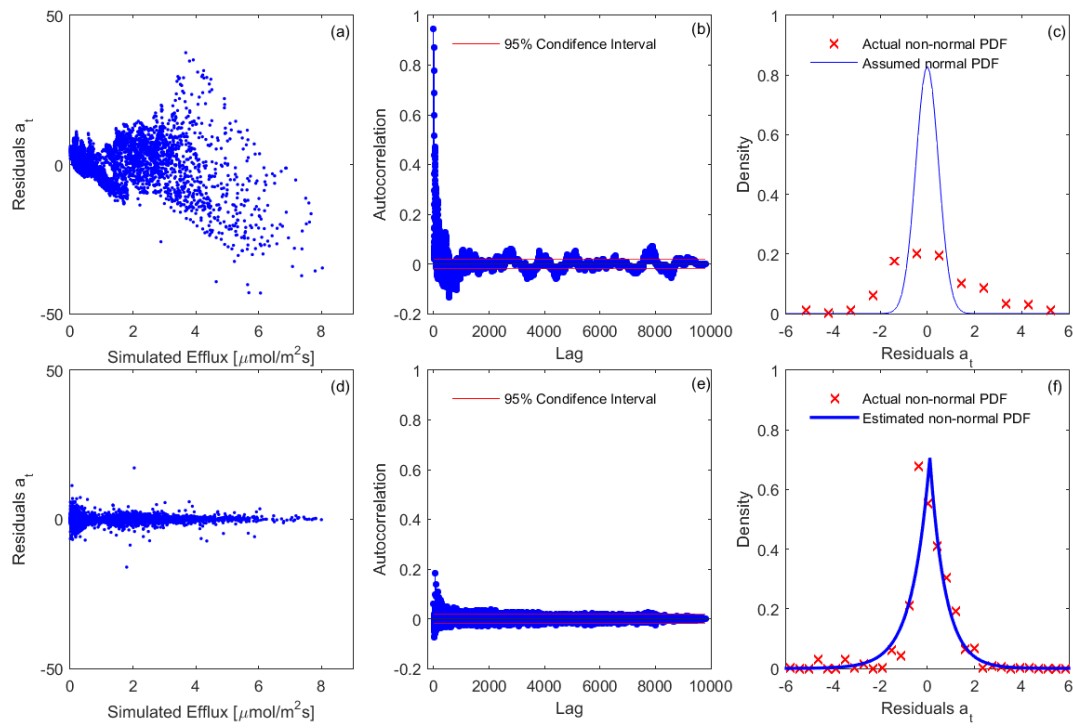






Figure 4. Residual quantile-quantile (Q-Q) plots of the best realization (among multiple MCMC
realizations) for the three soil respiration models and eight data models.

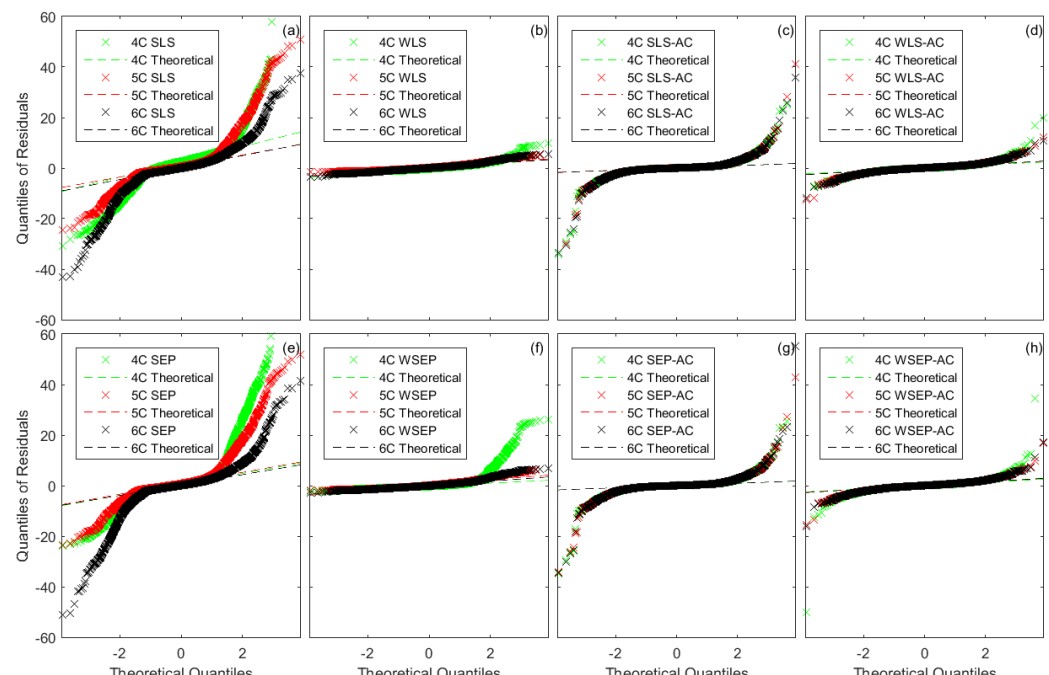






Figure 5. Marginal posterior parameter density of carbon use efficiency (CUE) for the three soil
respiration models and eight data models.

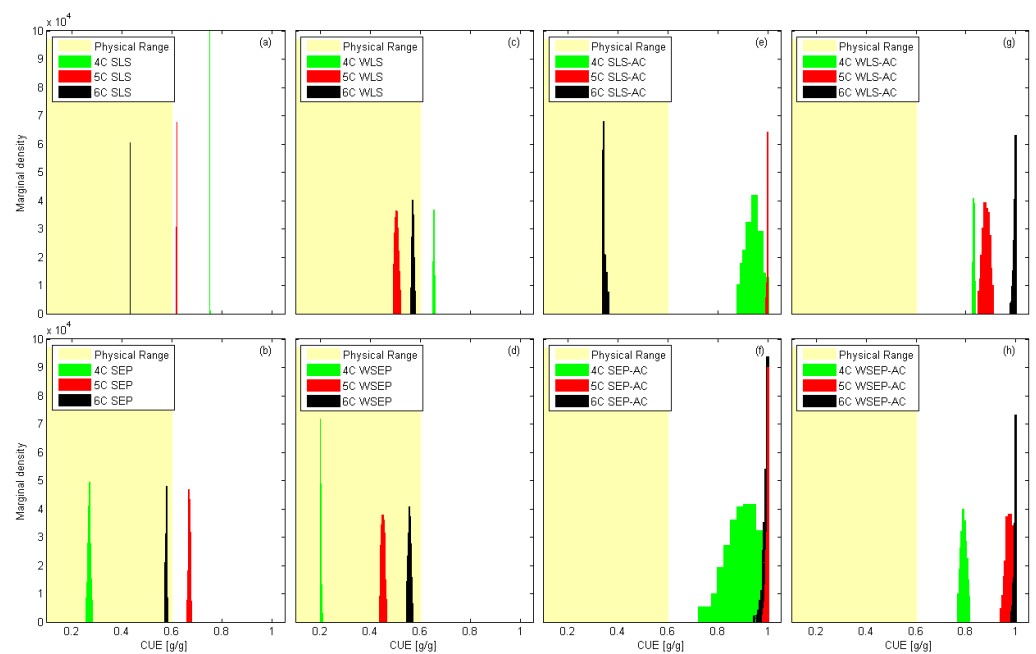




Figure 6. Observation data (blue dots) and mean prediction (green line) and 95% credible intervals
(red line) of prediction ensembles for (a)-(f) the calibration period and (g)-(l) the validation period.
The plots are for the three soil respiration models using data models SLS and WSEP-AC. *The
prediction ensembles are generated to consider parametric uncertainty of the soil respiration
models only.*



Figure 7. (a-b) Nash-Sutcliffe model efficiency (NSME), (c)-(d) sharpness, (e)-(f) predictive
coverage, and (g)-(h) relative model score for measuring predictive performance of the three soil
respiration models and the eight data models during the calibration and cross-validation periods.
*The statistics are evaluated from the prediction ensembles generated to consider parametric*
*uncertainty of the soil respiration models only.*

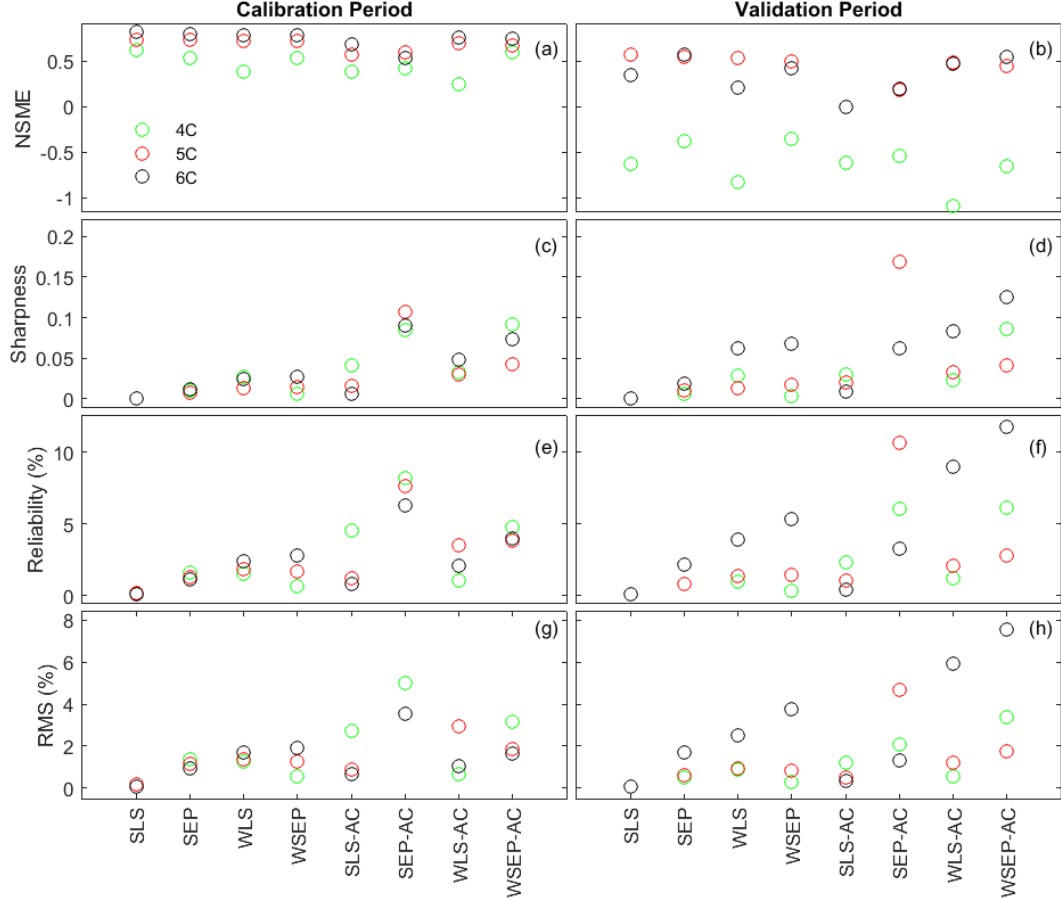





Figure 8. Observation data (blue dots) and mean prediction (green line) and 95% credible intervals
(red line) of prediction ensembles for (a)-(f) the calibration period and (g)-(l) the validation period.
The plots are for the three soil respiration models using data models SLS and WSEP-AC. *The*
*prediction ensembles are generated to consider parametric uncertainty of not only the soil*
*respiration models but also the data models.*

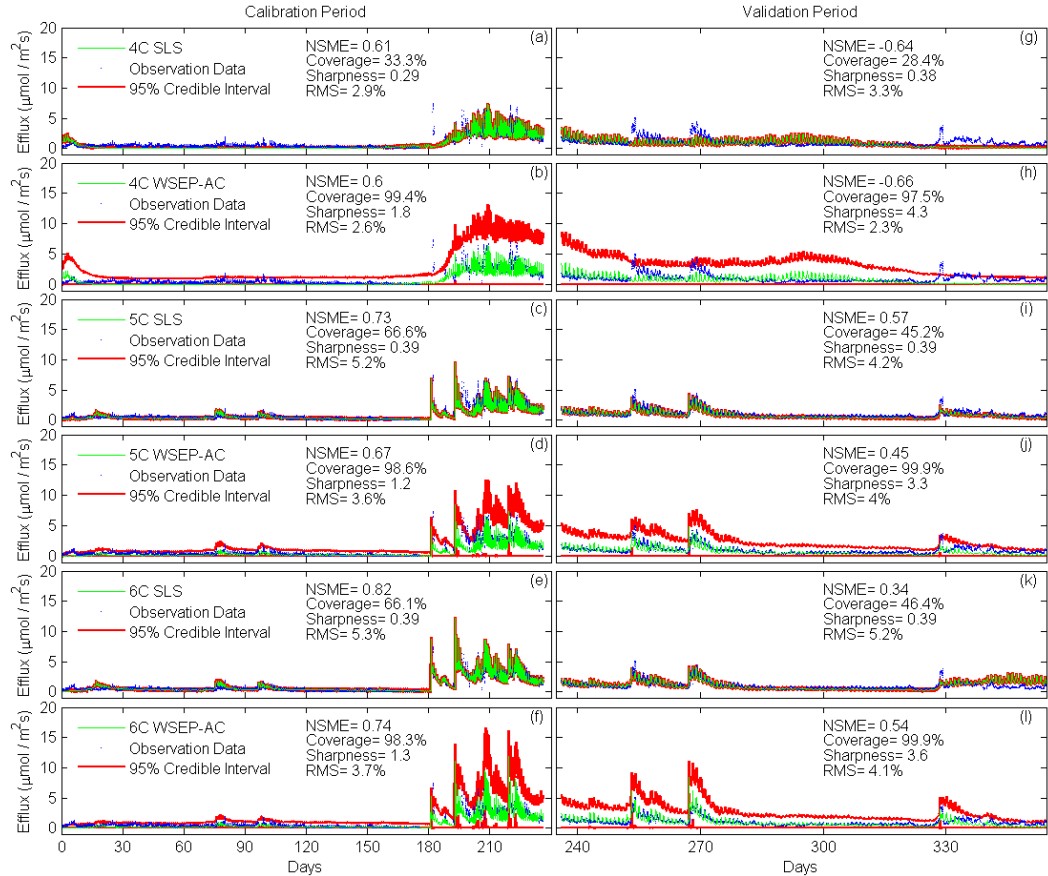






Figure 9. (a-b) Nash-Sutcliffe model efficiency (NSME), (c)-(d) sharpness, (e)-(f) predictive coverage, and (g)-(h) relative model score for measuring predictive performance of the three soil respiration models and the eight data models during the calibration and cross-validation periods. *The statistics are evaluated from the prediction ensembles generated to consider parametric uncertainty of not only the soil respiration models but also the data models.*

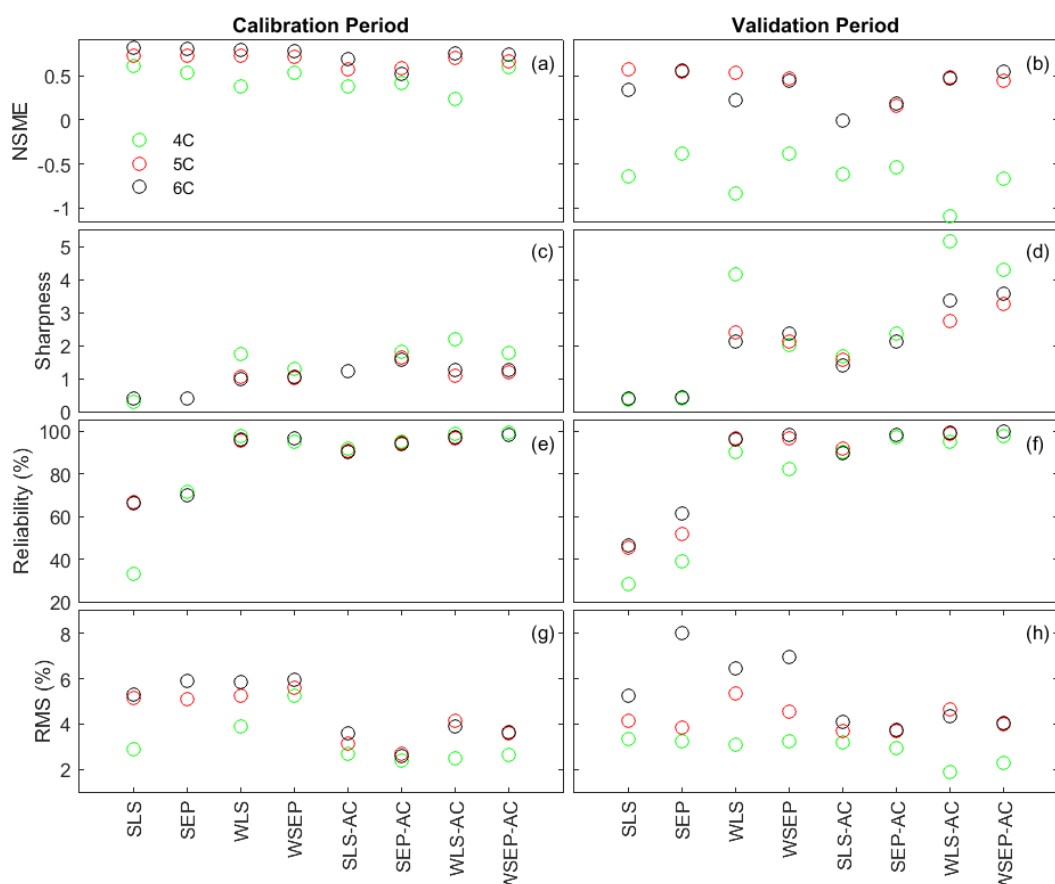





Figure 10. Observation data (blue dots) and mean prediction (green line) and 95% credible intervals (red line) for 6C for the eight likelihood functions during the calibration period (a)-(h) and the validation period (i)-(p). *The prediction ensembles are generated to consider parametric uncertainty of not only the soil respiration models but also the data models.*