# Peer review of "Bayesian Inference and Predictive Performance of Soil Respiration Models in the Presence 2 of Model Discrepancy"

_Geoscientific Model Development, 2018_

## Referee Comment (RC1) · Anonymous Referee #1 · 17 Jan 2019

The paper evaluates the impacts of statistical data assumptions in soil microbial respiration modeling on estimated model parameters and on model predictions. Inference is done using various soil respiration models and various likelihood functions, using half-hourly CO2 flux data from a field site. It's an interesting study, but I suggest additional effort to clarify and increase contribution of the work.

1. Contribution: the authors should more clearly spell out the explicit contributions of the paper. On the one hand, the methodology is not new and has been developed and applied in hydrological studies. On the other hand, the application to CO2 modeling may also not be entirely new since the likelihood approach used here has already

been applied to ecological modeling (including carbon flux modeling); a recent example is Scholz, K., Hammerle, A., Hiltbrunner, E. et al. Ecosystems (2018) 21: 982. https://doi.org/10.1007/s10021-017-0201-5.

2. The authors find some problems with the estimation of autocorrelation and suggest an alternative approach (Evin et al.). Why not test this approach as well? I'm not sure this would warrant a separate publication. Including it here would enhance novelty of the paper in my opinion. Note also that the high temporal resolution (half hourly) of the data used by the authors may be a complicating factor; see the following paper that discusses this: https://www.hydrol-earth-syst-sci-discuss.net/hess-2018-406/.

3. The paper should be checked for various grammatical errors and typos. One example is "heteroscedasticity", which is spelled in multiple creative ways throughout the paper.

4. Description of the various evaluation metrics seems better placed in the methods than results section.

5. Terminology: the distinction between model fidelity and discrepancy is not clear

6. Line 305, "discrete proposal distribution": I don't think the proposal is discrete, it is a proposal distribution over a continuous parameter space.

7. Line 477: please rephrase; I don't think it's "expected" that accounting for autocorrelation leads to biased parameter values. I would expect the opposite, since autocorrelation provides a (simple) way to account for model errors.

8. Eq. 23: is index i an index over time or is it an ensemble index? Please clarify.

9. Line 598: approaches that use "total residual error" typically still separate out parametric uncertainty, so the residual error includes measurement, model input, and model structure uncertainty, but not parameter uncertainty.

---

## Referee Comment (RC2) · Anonymous Referee #2 · 30 Jan 2019

The manuscript submitted by Elshall et al. is an interesting study dealing with the complexity of soil C model parameterization. In recent decades, the complexity of those model as well as the different tools to parameterize has increased substantially leading to potential misuses of powerful but complex mathematical approaches. The goal of Elshall et al is therefore to evaluate the impact on process-based model predictions of neglecting a couple of assumptions of the Bayesian framework as it is often done by soil modelers to avoid complexity.

The present study might not be super novel for the entire modeling communities in geoscience as mentioned by the other referee. Nevertheless, it underlines a flaw of

several carbon soil modeling studies and might be considered as novel in this context. It is a pity that the author may not freely communicate their models and scripts it would have definitely increased the impact of the paper.

Even though the objectives of the paper are important and deserve to be published, in my opinion, the manuscript in its present form is sometimes too hard to read and needs some simplifications. A first recommendation might be to have a table summarizing all the acronyms and try to reduce them when not necessary. Secondly, a workflow scheme might also be useful to understand the logic of the authors, which is not always super clear. Finally, I missed some definition to be sure I fully understood the text. In particular, it is not crystal clear to me what the author means by 'data model'. From my understanding, a data model is based on data but the observed data are presented quite fare from the data model. Another point is that I still do not fully understood how the authors link their data model with their process-based model. I understood that the data models are used for posterior parameter estimation but sometimes the text makes me doubt.

I don't understand why the author fixed the upper limit of the physical range of CUE to 0.6 (the mean over terrestrial systems) whereas in the paper they cited several observations are above 0.6

Some typo: l121 'and' not necessary L176 please correct the parenthesis L611: despite instead of desp8ite

I, therefore, think that this manuscript deserves publication after a deep rewriting to clarify the methods used.
* * *

---

## Author Comment (AC1) · 14 Mar 2019

Comment: The paper evaluates the impacts of statistical data assumptions in soil microbial respiration modeling on estimated model parameters and on model predictions. Inference is done using various soil respiration models and various likelihood functions, using half hourly $CO_2$ flux data from a field site. It's an interesting study, but I suggest additional effort to clarify and increase contribution of the work.

Response: We are very thankful for the reviewer for talking the time to evaluate the

manuscript, and for providing constructive comments.

Comment: 1. Contribution: the authors should more clearly spell out the explicit contributions of the paper. On the one hand, the methodology is not new and has been developed and applied in hydrological studies. On the other hand, the application to CO2 modeling may also not be entirely new since the likelihood approach used here has already been applied to ecological modeling (including carbon flux modeling); a recent example is Scholz, K., Hammerle, A., Hiltbrunner, E. et al. Ecosystems (2018) 21: 982. https://doi.org/10.1007/s10021-017-0201-5.

Response: We explicitly spelled out the novel contrition of this paper, which is the systematic evaluation of the impact of data model selection on Bayesian inference and predictive performance of soil respiration modeling with different degrees of model fidelity. We did a systematic review of Bayesian inference for soil respiration modeling. Most studies assume independent, Gaussian, and homoscedastic residuals. Few studies have relaxed these assumptions(e.g. Elshall et al., 2018; Scholz et al. 2018). However, only very few studies have focused on investigating the impacts of these assumptions for soil respiration modeling by relaxing the independent residuals assumption (Ricciuto et al., 2011) and the Gaussian residuals assumption (Ricciuto et al., 2011; van Wijk et al., 2008). By relaxing these three assumptions step-wise resulting in eight data models, to our knowledge this is the first study that systematically evaluates the impact of data models on Bayesian inference and predictive performance of soil respiration modeling.

The revised manuscript reads: "Bayesian inference of soil respiration models often adopts the assumption of independent, normally distributed and homoscedastic residuals (e.g. Ahrens et al., 2014; Bagnara et al., 2015, 2018; Barr et al., 2013; Barrongafford et al., 2014; Braakhekke et al., 2014; Braswell et al., 2015; Correia et al., 2012; Du et al., 2015, 2017; Hararuk et al., 2014; Hashimoto et al., 2011; He et al., 2018; Klemedtsson et al., 2008; Menichetti et al., 2016; Raich et al., 2002; Ren et al., 2013; Richardson and Hollinger, 2005; Steinacher and Joos, 2016; Tucker et al., 2014; Tuomi

et al., 2008; Xu et al., 2006; Yeluripati et al., 2009; Yuan et al., 2012, 2016; Zhang et al., 2014; Zhou et al., 2010). These assumptions are conveniently adopted since the requirement of using an unknown probability model in Bayesian statistics is called "a basic dilemma" by Box and Tiao (1992). Postulating the data models is always based on assumptions about residual statistics, and the most widely used assumptions are paired as follows: (i) independent vs. correlated residuals, (ii) homoscedastic vs. heteroscedastic residuals, and (iii) Gaussian vs. non-Gaussian residuals. For soil respiration modeling few studies have relaxed the independent residuals assumption (e.g. Cable et al., 2008, 2011; Li et al., 2016b), the homoscedasticity assumption (e.g. Berryman et al., 2018; Elshall et al., 2018; Ogle et al., 2016; Tucker et al., 2013), and the non-Gaussian and homoscedasticity assumptions (e.g. Elshall et al., 2018; Ishikura et al., 2017; Kim et al., 2014). A recent study (Scholz et al., 2018) relaxed these three assumptions using the generalized likelihood function (Schoups and Vrugt, 2010). However, few studies have focused on investigating appropriateness and impact of these assumptions for soil respiration modeling. This was performed by relaxing the independent residuals assumption (Ricciuto et al., 2011) and the Gaussian residuals assumption (Ricciuto et al., 2011; van Wijk et al., 2008). By relaxing these three assumptions stepwise resulting in eight data models, to our knowledge this is the first study that systematically evaluates the impact of data model selection on Bayesian inference and predictive performance of soil respiration modeling. In addition, to our knowledge this is the first soil respiration modeling study that investigates the impact of data models in relation to model fidelity." In the first paragraph of the introduction we also stated "While a large number of data models have been used (e.g. Elshall et al., 2018; Scholz et al., 2018) to our knowledge comprehensive and systematic evaluation of data models for soil respiration modeling has not been reported in literature."

Comment: 2. The authors find some problems with the estimation of autocorrelation and suggest an alternative approach (Evin et al.). Why not test this approach as well? I'm not sure this would warrant a separate publication. Including it here would enhance novelty of the paper in my opinion. Note also that the high temporal resolution (half hourly) of the data used by the authors may be a complicating factor; see the following paper that discusses this: https://www.hydrol-earth-syst-sci-discuss.net/hess-2018-406/.

Response: Thank you very much for bring our attention to this recent article of Ammann et al. (2018).

This manuscript provides a systematic evaluation of the impact of data model selection on Bayesian inference and predictive performance of soil respiration modeling. Figure 10 for example shows specific trends that would occur when relaxing the three assumptions of non-correlation, normality, and homoscedasticity using joint inversion approach, which has never been reported before in literature.

Autocorrelation is a complicated problem that we are currently working on. Joint inversion of heteroscedasticity and autocorrelation parameters can lead to poor predictive performance (Evin et al., 2013, 2014; Ammann et al. 2018; and this study). To address this problem a two-step procedure (e.g. Lu et al., 2013; Evin et al., 2013, 2014) was proposed. Our preliminary results show that using the sequential approach of Evin et al. (2013; 2014) by estimating the autoregressive parameters sequentially (after estimating the soil respiration model parameters and data-model parameters) did not solve this problem. Ammann et al. (2018) even states that the joint inversion is still preferred, and understanding the conditions where accounting for auto-correlation can be achieved remain poorly understood. The problem of autocorrelation has several interlinked aspects that we would like to address in another manuscript. Auto-correlated errors might be attributed to a systematic error in the soil respiration model. The most obvious solution is to improve the soil respiration model. Otherwise, we can improve our data model. Our hypothesis that we would like to test is that omitting autocorrelation error through a filter approach (e.g. Schoups and Vrugt, 2010; Evin et al., 2013; 2014; this study) could be tricky as this leads to a loss of information content. Thus, joint approach may lead to biased parameter estimation (Figure 5) and poor predictive performance (Figure 10). While sequential approach would avoid the biased parameter

estimation, but would still lead a poor predicative performance.

Our current understanding is that this problem could emerge from several interlinked factors: • Non-stationarity due to wet-dry periods as proposed by Ammann et al. (2018) could be a reason for this problem and thus accounting for non-stationarity (Smith et al., 2010b, Ammann et al. 2018) could alleviate this problem. • The method for accounting for autocorrelation could have an impact. Autocorrelation could be addressed using a likelihood function based on covariance matrix of residuals L(e) (e.g. Lu et al., 2013) with transformed residuals, and likelihood function of normalized residuals L(a) (e.g. Schoups and Vrugt, 2010; Evin et al., 2013; 2014; this study) with autoregressive model that filter out autocorrelation. Note that "e" is a vector of transformed residuals, while "a" is a vector independent and identically distributed random errors with zero mean and unit standard deviation. The impact of method selection is still unclear and needs investigation. • Joint versus sequential inversion for autocorrelation could also have an impact. Ammann et al. (2018) suggests that the joint inversion is still preferred over sequential inversion. This will be investigated under both L(e) and L(a) approaches. In addition, we would like to test novel joint inversion procedure that combines the L(a) and L(e) approaches as follows. First, the parameters of the linear heteroscedastic model will be estimated similar to Schoups and Vrugt (2010) to remove heteroscedasticity. For each MCMC sample, after applying the linear heteroscedasticity model, the auto-correlation parameters can be deterministically calculated as internal variables of the data model similar to Lu et al. (2013) and not as calibration parameters(e.g. Schoups and Vrugt, 2010; Evin et al., 2013;2014). This is mainly to avoid interaction between heteroscedasticity and autocorrelation parameters. The auto-correlation parameters can be calculated following Lu et al. (2013). All these interlinked factors require careful consideration, and this is warranted in another manuscript.

We have revised the manuscript to further clarify these issues as follows: "This study confirms the empirical findings and theoretical analysis (Evin et al., 2013; 2014; Am-

mann et al. 2018) that separate accounting for autocorrelation or joint inversion of correlation and heteroscedasticity can be problematic. By drawing on similarity from surface hydrology, the study of Ammann et al. (2018) suggests that this might be attributed to non-stationarity due to wet-dry periods with half-hourly data. Accounting for non-stationarity (Smith et al., 2010b, Ammann et al. 2018) could address this problem. Relatively poor performance with respect to autocorrelation can be also attributed to the implementation scheme. The inference scheme such as joint inference as in this study, post-processing inference approach for autocorrelation (Evin et al., 2013; 2014), residuals transformation approach (e.g. Lu et al., 2013) or other strategies (Li et al., 2015, 2016a) could have an impact. Yet Ammann et al., (2018) study states that the joint inversion is still preferred, and understanding the conditions where accounting for auto-correlation can be achieved remain poorly understood. Further investigation of this point is warranted in a future study."

Comment: 3. The paper should be checked for various grammatical errors and typos. One example is "heteroscedasticity", which is spelled in multiple creative ways throughout the paper.

Response: Thank you very for pointing this out and we have corrected "heteroscedasticity" at eight different locations throughout the manuscript. We corrected several other grammatical errors and typos.

Comment: 4. Description of the various evaluation metrics seems better placed in the methods than results section.

Response: We moved the description of the various evaluation metrics from the results to the methods section.

Comment: 5. Terminology: the distinction between model fidelity and discrepancy is not clear

Response: We clarified these two terms as follows: "We use the terms model fidelity

and model discrepancy interchangeably. Model fidelity refers to the degree of realism of representing our scientific knowledge with respect to the real world system. That is a high fidelity model has less discrepancy."

Comment: 6. Line 305, "discrete proposal distribution": I don't think the proposal is discrete, it is a proposal distribution over a continuous parameter space.

Response: We revised "discrete proposal distribution" to "adaptive proposal distribution."

Comment: 7. Line 477: please rephrase; I don't think it's "expected" that accounting for autocorrelation leads to biased parameter values. I would expect the opposite, since autocorrelation provides a (simple) way to account for model errors.

Response: We rephrased this sentence to "First, we obtained biased parameter estimates that is out the reasonable physical range."

Comment: 8. Eq. 23: is index i an index over time or is it an ensemble index? Please clarify.

Response: Thank you very much for point this out. We clarified that this is an ensemble prediction Yij where i is index over time, and revised other parts of the manuscript accordingly. The new sentence read "the ensemble prediction Yij is similar to Yi above where is index over time and specific to the j-th combination."

Comment: 9. Line 598: approaches that use "total residual error" typically still separate out parametric uncertainty, so the residual error includes measurement, model input, and model structure uncertainty, but not parameter uncertainty.

Response: That is true. We rephrased that sentence to "total residuals that separates out parametric uncertainty, so the residual error includes measurement, model input, and model structure uncertainty."

Thank you very much for your constructive comments.

---

## Author Comment (AC2) · 14 Mar 2019

Comment: The manuscript submitted by Elshall et al. is an interesting study dealing with the complexity of soil C model parameterization. In recent decades, the complexity of those model as well as the different tools to parameterize has increased substantially leading to potential misuses of powerful but complex mathematical approaches. The goal of Elshall et al is therefore to evaluate the impact on process-based model predictions of neglecting a couple of assumptions of the Bayesian framework as it is often done by soil modelers to avoid complexity.

[Figure]

Response: We thank the reviewer very much evaluating the manuscript and for providing constructive feedback and suggestions.

Comment: The present study might not be super novel for the entire modeling communities in geoscience as mentioned by the other referee. Nevertheless, it underlines a flaw of several carbon soil modeling studies and might be considered as novel in this context. It is a pity that the author may not freely communicate their models and scripts it would have definitely increased the impact of the paper.

Response: We feel sorry for this too, and we would love to share the code and the soil respiration models upon request.

Comment: Even though the objectives of the paper are important and deserve to be published, in my opinion, the manuscript in its present form is sometimes too hard to read and needs some simplifications. A first recommendation might be to have a table summarizing all the acronyms and try to reduce them when not necessary.

Response: We added a list of acronyms as follows:

Acronyms 4C Four carbon pool model 5C Five carbon pool model 6C Six carbon pool model CUE Microbial carbon use efficiency DOC Dissolved organic carbon ENZ Enzymes MCMC Markov chain Monte Carlo MIC Microbial biomass NSME Nash-Sutcliffe model efficiency PDF Probability density function RMS Relative model score SEP Skew exponential power distribution SEP-AC Skew exponential power distribution with autocorrelation SLS Standard least square SLS-AC Standard least square with autocorrelation SOC Soil organic carbon WLS Weighted least squared WLS-AC Weight least square with autocorrelation WSEP Weighted skew exponential power distribution WSEP-AC Weighted skew exponential power distribution with autocorrelation

Comment: Secondly, a workflow scheme might also be useful to understand the logic of the authors, which is not always super clear.

Response: We added a summary table of the data models and corresponding likeli-

hood functions. The revised manuscripts states "A summary table of the eight data models with corresponding parameters is provided in the supplementary materials." We added a workflow scheme as a supplementary figure. The revised manuscript reads " Our workflow scheme is presented in the supplementary materials." The new table and figure are present in the attached supplementary pdf file.

Comment: Finally, I missed some definition to be sure I fully understood the text. In particular, it is not crystal clear to me what the author means by 'data model'. From my understanding, a data model is based on data but the observed data are presented quite fare from the data model.

Response: In the revised manuscript we clarified that "A data model that is also known as a residuals model or an error model is used to characterize residuals (i.e., the difference between data and corresponding model simulations)." In addition, please see our response to the previous comment.

Comment: Another point is that I still do not fully understood how the authors link their data model with their process-based model. I understood that the data models are used for posterior parameter estimation but sometimes the text makes me doubt.

Response: The parameters of the data model are jointly estimated with the parameters of the soil respiration model using MCMC. We clarified this in the revised manuscript "the posterior distributions of the data model parameters are jointly estimated with the soil respiration model parameters using the MT-DREAM(ZS) code (Laloy and Vrugt, 2012)." In addition, a summary of the data model parameters is presented in the supplementary materials as we clarified in a previous response.

Comment: I don't understand why the author fixed the upper limit of the physical range of CUE to 0.6 (the mean over terrestrial systems) whereas in the paper they cited several observations are above 0.6

Response: The thermodynamic maximum limit of CUE is 0.6 and the empirical observations show that CUE over a wide range of field conditions converges to $\sim 0.30$ with a mean value of 0.55 for terrestrial ecosystems (Sinsabaugh et al., 2013). We used this upper limit for analysis only. We did not fix this limit for Bayesian inverse modeling to understand the impact of data model on parameter estimation.

Comment: Some typo: l121 'and' not necessary L176 please correct the parenthesis L611: despite instead of desp8ite

Response: Thank you very much for pointing out these typos and we corrected them. Thank you very much.

Comment: I, therefore, think that this manuscript deserves publication after a deep rewriting to clarify the methods used

Response: Addressing the review comments helped us to rewrite and clarify several parts of the manuscript. Thank you very much.

Please also note the supplement to this comment:
https://www.geosci-model-dev-discuss.net/gmd-2018-272/gmd-2018-272-AC2-supplement.pdf

———————————————

[Figure]

**Supplement:**

Supplementary Table 1. Summary of the data models and corresponding likelihood functions

| Residual Assumptions | Likelihood function | Data model | Residuals | Variance | Likelihood function parameters |
|---|---|---|---|---|---|
| | | Generic data model $$a_t = \frac{\varepsilon_t}{\sigma_t} \quad a_t \sim X$$ | $\varepsilon_t$ | $\sigma_t$ | |
| Independent, normally distributed, and homoscedastic | Standard least square (SLS) | $$a_t = \frac{\varepsilon_t}{\sigma_0} \quad a_t \sim N(0,1)$$ | $\varepsilon_t = d_t - Y_t$ | $\sigma_t = \sigma_0$ | Constant $\sigma_0$ |
| Independent, and homoscedastic | Skew exponential power (SEP) | $$a_t = \frac{\varepsilon_t}{\sigma_0} \quad a_t \sim SEP(0,1,\xi,\beta)$$ | $\varepsilon_t = d_t - Y_t$ | $\sigma_t = \sigma_0$ | Constant $\sigma_0$ Skewness $\xi$, Kurtosis $\beta$ |
| Independent and normally distributed | Weighted least square (WLS) | $$a_t = \frac{\varepsilon_t}{\sigma_0 + \sigma_1 Y_t} \quad a_t \sim N(0,1)$$ | $\varepsilon_t = d_t - Y_t$ | $\sigma_t = \sigma_0 + \sigma_1 Y_t$ | Heteroscedasticity model parameters $\sigma_0, \sigma_1$ |
| Independent | Weighted skew exponential power (WSEP) | $$a_t = \frac{\varepsilon_t}{\sigma_0 + \sigma_1 Y_t} \quad a_t \sim SEP(0,1,\xi,\beta)$$ | $\varepsilon_t = d_t - Y_t$ | $\sigma_t = \sigma_0 + \sigma_1 Y_t$ | Heteroscedasticity model parameters $\sigma_0, \sigma_1$ Skewness $\xi$, Kurtosis $\beta$ |
| Normally distributed, and homoscedastic | Standard least square with auto-correlation (SLS-AC) | $$a_t = \frac{\varepsilon_t - \sum_{i=1}^{p} \phi_i \varepsilon_{t-i}}{\sigma_0} \quad a_t \sim N(0,1)$$ | $\varepsilon_t - \sum_{i=1}^{p} \phi_i \varepsilon_{t-i}$ | $\sigma_t = \sigma_0$ | Constant $\sigma_0$, Autoregressive model parameters $\phi_i$ |
| Homoscedastic | Skew exponential power with auto-correlation (SEP-AC) | $$a_t = \frac{\varepsilon_t - \sum_{i=1}^{p} \phi_i \varepsilon_{t-1}}{\sigma_0} \quad a_t \sim SEP(0,1,\xi,\beta)$$ | $\varepsilon_t - \sum_{i=1}^{p} \phi_i \varepsilon_{t-i}$ | $\sigma_t = \sigma_0$ | Constant $\sigma_0$, Autoregressive model parameters $\phi_i$ Skewness $\xi$, Kurtosis $\beta$ |
| Normally distributed | Weighted least square with auto-correlation (WLS-AC) | $$a_t = \frac{\varepsilon_t - \sum_{i=1}^{p} \phi_i \varepsilon_{t-1}}{\sigma_0 + \sigma_1 Y_t} \quad a_t \sim N(0,1)$$ | $\varepsilon_t - \sum_{i=1}^{p} \phi_i \varepsilon_{t-i}$ | $\sigma_t = \sigma_0 + \sigma_1 Y_t$ | Heteroscedasticity model parameters $\sigma_0, \sigma_1$ Autoregressive model parameters $\phi_i$ |
| | Generalized likelihood function (WSEP-AC) | $$a_t = \frac{\varepsilon_t - \sum_{i=1}^{p} \phi_i \varepsilon_{t-1}}{\sigma_0 + \sigma_1 Y_t} \quad a_t \sim SEP(0,1,\xi,\beta)$$ | $\varepsilon_t - \sum_{i=1}^{p} \phi_i \varepsilon_{t-i}$ | $\sigma_t = \sigma_0 + \sigma_1 Y_t$ | Heteroscedasticity model parameters $\sigma_0, \sigma_1$ Autoregressive model parameters $\phi_i$, Skewness $\xi$, Kurtosis $\beta$ |

Supplementary Figure 1. Workflow scheme

**Bayesian Inference**

**Data Model and Likelihood Function**
- Formulate a data model by stepwise relaxing the independent, normally distributed and homoscedastic residual assumptions
- Given a data model, formulate likelihood function SLS, WLS, SEP, WSEP, SLS-AC, WLS-AC, SEP-AC or WSEP-AC

**Bayesian Inverse Modeling**
- Select a soil respiration model: 4C,5C or 6C
- Specify the prior parameter distributions for the soil respiration model and the likelihood function
- Run MCMC to update the prior to posterior parameter distributions given a likelihood function

**Residuals Characterization**
- Plot residuals and quantile-quantile (Q-Q) plot to understand the validity of the residual assumptions

**Posterior Parameter Distributions**
- Analyze the impact of the eight data model on parameter estimation

**Impact of Model Discrepancy**
- Use the three soil respiration models to study the impact of model discrepancy on residuals and parameter estimation given eight data models

**Model Prediction**

**Credible Interval**
- Sample posterior distributions of the soil respiration model to generate the prediction ensemble given parametric uncertainty of the soil respiration model

**Predictive Interval**
- Sample posterior distributions of the soil respiration model and data model to generate the prediction ensemble given total uncertainty

**Predictive Performance Evaluation**
- Evaluate the predictive performance with respect to central mean tendency using Nash-Sutcliffe model efficiency, dispersion using sharpness metric, and reliability using predictive coverage
- Evaluate the overall predictive performance using the scoring rule of relative model score

**Impact of Model Discrepancy**
- Evaluate the predictive performance for the three soil respiration models to study the impact of model discrepancy given eight data models

---

## Author Response (AR2)

**Bold black font:** **Topical editor and reviewer comments**
Black font: Author response
Blue font: Verbatim copy and paste from the revised manuscript

**Topical Editor Decision: Publish subject to minor revisions (review by editor) (08 Apr 2019) by Christoph Müller**

**Comments to the Author:**
**Dear Dr. Elshall,**

**the original reviewers have seen your revised paper again and suggest further amendments to the paper. I think what they suggest is easy enough to implement.**

**I look forward to your revised paper.**

**Cheers**
**Christoph**

Thank you very much for handling the original and the revised submissions. We have responded to all the Editorial Support and reviewers' comments, and revised the manuscript accordingly.

**Editorial Support**

**Besides adjustments requested by the Topical Editor or Referees, please check your manuscript carefully for typos, missing co-authors and their affiliations, terminology, updates of data in tables, or updates of variables in equations.**

We updated the affiliation of the second author and the funding information. We improved the article keywords. We carefully checked the manuscript, and corrected several typos and grammatical errors, as shown in the *marked-up manuscript version.* We also slightly improved the writing style in several parts of the article and added more clarifications as shown in the *marked-up manuscript version.*

**Anonymous Referee #1**

**I revisit some of my previous comments, considering response from the authors in the discussion forum.**

**1. Contribution: the authors have now better articulated their contribution. The paper is somewhat incremental, since the methodology is not new and the findings are similar to applications in other disciplines. Nevertheless it is still useful to have an explicit evaluation and comparison of the impacts of statistical assumptions in soil respiration models.**

Thank you for taking the time to review the manuscript and for your valuable feedback.

**2. Problems with residual autocorrelation: I agree that the authors do not need to solve this issue in this paper. However, the alternative suggested approach (Evin et al) could be easily tested (simply swap the order in which correlation and heteroscedasticity are applied in the likelihood function). In their extended response in the discussion forum the authors seem to indicate that they have tested this. Why then not add it to the paper? If not in the results then in the discussion part of the paper ("preliminary**

**results show...").** My suggestion is to include the entire extended response posted in the discussion forum (under comment 2 of RC1) in the paper itself, as it provides more information that is useful for readers.

We prefer to keep the manuscript in its current form since we are afraid that we cannot adequately address this autocorrelation problem in this manuscript. We prefer not to present incomplete or preliminary results in a published paper. However, we improved the extended response, and add it to the manuscript as suggested by the reviewer. The added part reads:

**4.3 Discussion on handling residual correlation**

Accounting for autocorrelation can lead to biased parameter estimation (Figure 5) and poor predictive performance (Figure 10). Auto-correlated residuals may be attributed to model discrepancy, as shown in Lu et al. (2013). The most obvious solution to handle the autocorrelation is to reduce the autocorrelation by improving the soil respiration model. If model improvement is difficult for practical reasons, we can improve the data model to better characterize the autocorrelation. Addressing autocorrelation in a data model is challenging since it involves several interlinked factors as follows:

(1) Non-stationarity due to wet-dry periods could be a reason for this problem. By drawing on similarity from surface hydrology, the study of Ammann et al. (2018) suggests that auto-correlated residuals might be attributed to non-stationarity due to wet-dry periods with half-hourly data. Accounting for non-stationarity could better address the problem of auto-correlated residuals (Ammann et al., 2018; Smith et al., 2010b).

(2) The way of implementing autocorrelation could have an impact. Autocorrelation could be applied to raw residuals directly (e.g., Li et al., 2015), to transformed residuals based on covariance matrix of residuals L($e$) (e.g., Lu et al., 2013), or to normalized residuals L($a$) (e.g., Schoups and Vrugt, 2010; Evin et al., 2013). Note that $e$ is a vector of transformed residuals, while $a$ denotes a vector of independent and identically distributed random errors with zero mean and unit standard deviation. The L($e$) approach based on covariance matrix of residuals is generally limited to Gaussian data models (e.g. Lu et al., 2013), while the L($a$) approach for normalized residuals can be readily adopted for non-Gaussian data models.

(3) The autocorrelation model could have an impact. Using an autoregressive model is a popular technique to account for auto-correlated residuals. However, using an autoregressive model with either joint inversion approach (e.g., this study and Schoups and Vrugt, 2010) or sequential approaches (e.g., Evin et al., 2013, 2014; Lu et al., 2013) removes correlation errors through a filter approach, which can lead to a loss of information content. As this may cause overcorrection of prediction especially at surge events, Li et al. (2015) developed a restricted autoregressive model to overcome this adverse effect. Other autocorrelation models include moving average model and mixed autoregressive-moving averaging model (Chatfield, 2004).

(4) Joint versus sequential inversion for autocorrelation could have an impact. Sequential inversion approaches include two-step procedures (e.g. Evin et al., 2013, 2014; Lu et al., 2013) or the multi-step procedure (Li et al., 2016a). These sequential approach estimates the autoregressive parameters sequentially in a later step after estimating the physical model parameters and other data model parameters. Evin et al. (2013, 2014) used a sequential approach to avoid the interaction between the parameters of the heteroscedasticity model and the autocorrelation model. In addition, the autoregressive model parameters can be deterministically calculated as an internal variables of the data model similar to Lu et al. (2013), and not as calibration parameters (e.g. Schoups and Vrugt; Evin et al. 2013; 2014). While the first step in the sequential approach would avoid the biased parameter estimation (Figure 10a-d), the second step can still lead a poor predicative performance since we are essentially using a filter approach to remove residual correlation. To address this problem, Li et al. (2016) multi-step procedure that is based on Gaussian data model uses restricted autoregressive model. Generally, Ammann et al. (2018) states that the joint inversion is still preferred, and understanding the conditions where accounting for auto-correlation can be achieved remain poorly understood.

In addition, the text about autocorrelation in the conclusions section was accordingly shortened. The revised manuscript reads "While the reasons remain poorly understood (Ammann et al., 2018), it might be attributed to non-stationarity due to wet-dry periods with half-hourly data (Ammann et al., 2018) or to the method of handling autocorrelation (e.g., Schoups and Vrugt, 2010, Evin et al., 2013; 2014; Lu et al., 2013; Li et al., 2015, 2016a; Ammann et al. 2018). Further investigation to address autocorrelation in soil respiration modeling is warranted in a future study."

**3. Grammatical/spelling errors: the authors state that they have "corrected several other grammatical errors", but it's not clear what was corrected exactly.**

Sorry for not listing the grammatical errors and typos the we corrected in the previous submission, which are as follows:
- Line 123: was used to select  the best model -> was used to select the best model
- Line 192: Laplace distribution  (van Wijk et al., 2008Ricciuto et al., 2011) -> Laplace distribution (van Wijk et al., 2008; Ricciuto et al., 2011).
- Line 207:  a dataset of CO2 -> by splitting the dataset of CO2
- Line 329: model parameters  jointly with -> model parameters jointly estimated with
- Line 420: each  -> each criterion
- Line 658: Accounting for ^ error term **e** -> Accounting for the error term **e**
- Line 688: the rest ^ six data models -> the rest of the six data models
- Line 722: the residuals, ^ thus resulting -> the residuals, and thus resulting
- Line 743: We tested eight data model -> We tested eight data models
- Line 807: The ^ conclusions  -> The above conclusions

**Additional comment:**
**- the abstract states that "not accounting for heteroscedasticity ... will definitely underestimate uncertainty". It is not clear why this would be the case. Perhaps you mean to say that not accounting for any residual error beyond parameter uncertainty will underestimate predictive uncertainty (as in section 4.2)? Although that is not really a significant finding.**

Comparing Figures 10a-b (with the SLS data model that does not account for residual error beyond parameter uncertainty) with Figures 10c-d (with the SEP data model that accounts for residual error beyond parameter uncertainty) shows that using an SEP data model with two additional parameters did not significantly impact the uncertainty. This is not the case for Figures 10e-h for data models WLS and WSEP that account for heteroscedasticity. Visual comparison of Figures 10a-d with Figures 10e-h and examination of the sharpness and predictive coverage metrics indicate that not accounting for heteroscedasticity will underestimate uncertainty. Accordingly, accounting for heteroscedasticity with WLS (Figures 10e-f) or WSELP (Figures 10g-h) makes the predictions more sensitive to peak carbon effluxes. We clarified this point, and the revised manuscript reads "Not accounting for heteroscedasticity will underestimate the predication uncertainty (Figure 10b and Figure 10d). This is mainly because the variance of the efflux residuals increases with the magnitude of the carbon effluxes (Figure 3a), and thus assuming constant variance is not representative. Accordingly, accounting for heteroscedasticity using WLS (Figure 10e) or WSEP (Figure 10h) will make the predictions more sensitive to peck carbon effluxes. This and will generally improve the predictive coverage on the expense of sharpness and the central mean tendency."

**Anonymous Referee #2**

**The revised version submitted by Elshall et al., generally answers quite well all my previous comment.**

Thank you for taking the time to review the original and the revise manuscripts.

**Nevertheless I still disagree with using a CUE maximum value of 0.6 and I would like to see some kind of sensitivity analysis to better understand the impact of such assumption. If the effect of changing the max CUE value is limited then the paper could be published in its current version. If not some discussions should be added.**

We probably did not make ourselves clear. For the inverse modeling with MCMC sampling we did not assume CUE maximum value of 0.6. Thus, for parameter estimation and predictive performance, we did not impose this constraint. We merely obtained this CUE maximum value of 0.6 from literature, which is based on thermodynamic calculation (Fernández-Martínez et al., 2014; Li et al., 2014; Sinsabaugh, et al., 2013), to evaluate whether the posterior parameter distributions of CUE under different data models and different soil respiration models are within this physically reasonable range of 0 ~ 0.6 or beyond. Thus, this assumption has no impact on the results. We clarified this in the revised manuscript as follows "Note that, for inverse modeling with MCMC sampling, we did not assume CUE maximum value of 0.6. In other words, for parameter estimation and predictive performance we did not impose the constraint that CUE is less than 0.6. We merely use this CUE maximum value of 0.6 to evaluate whether the posterior CUE parameter samples obtained using different data models and different soil respiration models are within the physically reasonable range of 0 ~ 0.6."

**Minor corrections:**
**L376 you mean soil moisture I guess?**
**L424 you mean Yi and not Xi, right?**

Thank you for these two typos and we corrected both of them.

[revised manuscript text omitted]

**Bayesian Inference**

**Data Model and Likelihood Function**
- Formulate a data model by stepwise relaxing the independent, normally distributed and homoscedastic residual assumptions
- Given a data model, formulate likelihood function SLS, WLS, SEP, WSEP, SLS-AC, WLS-AC, SEP-AC or WSEP-AC

**Bayesian Inverse Modeling**
- Select a soil respiration model: 4C,5C or 6C
- Specify the prior parameter distributions for the soil respiration model and the likelihood function
- Run MCMC to update the prior to posterior parameter distributions given a likelihood function

**Residuals Characterization**
- Plot residuals and quantile-quantile (Q-Q) plot to understand the validity of the residual assumptions

**Posterior Parameter Distributions**
- Analyze the impact of the eight data models on parameter estimation

**Impact of Model Discrepancy**
- Use the three soil respiration models to study the impact of model discrepancy on residuals and parameter estimation given eight data models

**Model Prediction**

**Credible Interval**
- Sample posterior distributions of the soil respiration model to generate the prediction ensemble given parametric uncertainty of the soil respiration model

**Predictive Interval**
- Sample posterior distributions of the soil respiration model and data model to generate the prediction ensemble given total uncertainty

**Predictive Performance Evaluation**
- Evaluate the predictive performance with respect to central mean tendency using Nash-Sutcliffe model efficiency, dispersion using sharpness metric, and reliability using predictive coverage
- Evaluate the overall predictive performance using the scoring rule of relative model score

**Impact of Model Discrepancy**
- Evaluate the predictive performance for the three soil respiration models to study the impact of model discrepancy given eight data models

---

## Author Response (AR3)

**Bold black font: Topical editor and reviewer comments**
Black font: Author response

**Topical Editor Decision: Publish subject to technical corrections (23 Apr 2019) by Christoph Müller**
**Comments to the Author:**
**Dear Dr. Elshall and co-authors,**

**thank you for your thorough revision of the manuscript.**
**I'm happy to accept the paper for publication.**
**There are two typos that need quick attention: "peck" should be "peak" in line 680 (and the response) and "soil inspiration models" in line 751 should be "soil respiration models" I suppose?**

**Thanks for submitting to GMD**

**Christoph**

Thank you very much for handling the manuscript and for accepting to publish our work in GMD. We made the following minor corrections:

(1) We corrected the typos in line 680 and line 751
(2) We corrected a grammatical error is line 745: "understanding the conditions where accounting for auto-correlation can be achieved remain^" → "understanding the conditions where accounting for auto-correlation can be achieved remains"
(3) We changed the marker color of the efflux observations in Figure 2 from green to blue to make it consistent with the other figures in the manuscript.

[revised manuscript text omitted]